# Somatic mutations in early metazoan genes disrupt regulatory links between unicellular and multicellular genes in cancer

Anna S Trigos[1,2], Richard B Pearson[2,3,4], Anthony T Papenfuss[1,2,5], David L Goode[1,2]*

[1]Computational Cancer Biology Program, Peter MacCallum Cancer Centre, Melbourne, Australia; [2]Sir Peter MacCallum Department of Oncology, The University of Melbourne, Parkville, Australia; [3]Department of Biochemistry and Molecular Biology, The University of Melbourne, Parkville, Australia; [4]Department of Biochemistry and Molecular Biology, Monash University, Clayton, Australia; [5]Bioinformatics Division, The Walter & Eliza Hall Institute of Medical Research, Parkville, Australia

**Abstract** Extensive transcriptional alterations are observed in cancer, many of which activate core biological processes established in unicellular organisms or suppress differentiation pathways formed in metazoans. Through rigorous, integrative analysis of genomics data from a range of solid tumors, we show many transcriptional changes in tumors are tied to mutations disrupting regulatory interactions between unicellular and multicellular genes within human gene regulatory networks (GRNs). Recurrent point mutations were enriched in regulator genes linking unicellular and multicellular subnetworks, while copy-number alterations affected downstream target genes in distinctly unicellular and multicellular regions of the GRN. Our results depict drivers of tumourigenesis as genes that created key regulatory links during the evolution of early multicellular life, whose dysfunction creates widespread dysregulation of primitive elements of the GRN. Several genes we identified as important in this process were associated with drug response, demonstrating the potential clinical value of our approach.
DOI: https://doi.org/10.7554/eLife.40947.001

*For correspondence:
david.goode@petermac.org

**Competing interests:** The authors declare that no competing interests exist.

## Introduction

The spectra of genomic and transcriptomic alterations across cancer types are highly heterogeneous. A multitude of driver genes exist both within and across subtypes (e.g. *Armenia et al., 2018*; *Nik-Zainal et al., 2016*) spanning a range of mutational types, from simple substitutions, to extensive genome-wide aneuploidies and complex rearrangements (*Ciriello et al., 2013*; *Hoadley et al., 2014*; *Kandoth et al., 2013*; *Lawrence et al., 2013b*; *Zack et al., 2013*). This complex array of genetic alterations confounds efforts to assign function to driver genes and identify key driver mutations within patients. Despite this extensive molecular heterogeneity, tumors originating from a variety of tissues converge to several common hallmark cellular phenotypes (*Hanahan and Weinberg, 2011*). Many of these involve loss of features commonly associated with multicellularity, for example, uninhibited proliferation, tissue dedifferentiation, disruption of cell-cell adhesion and intercellular communication, suggesting the alteration to genes involved in the evolution of multicellular traits is central to tumor development (*Aktipis et al., 2015*; *Aktipis and Nesse, 2013*; *Davies and Lineweaver, 2011*; *Vincent, 2012*).

**eLife digest** Cancers arise when harmful changes happen in the genetic information of certain cells. These 'mutations' are different from person to person, but overall, they disrupt healthy cells in similar ways. In particular, cancer cells tend to lose features that help cells work together in the body. Researchers have suggested that cancers may emerge when cells stop being able to cooperate with each other as part of an organism.

Our bodies still rely on old genes that were present in our earliest, single-cell ancestors. However, we also have newer genes that evolved when the organisms in our lineage started to have more than one cell. A complex network of signals exists to ensure that both sets of genes work together smoothly, and previous studies have suggested that cancers may appear when this delicate balance is disrupted.

To address this question, Trigos et al. have now used a computational approach to analyse different types of tumours from over 9,000 patients. This showed that, in cancer, many mutations disrupt the genes that coordinate old and new genes. These mutations were usually small, punctual changes in the genetic sequence. However, large modifications, such as an entire gene being deleted or repeated, took place more often in the old or the new genes themselves. Therefore, different classes of mutations have specific roles when disrupting how old and new genes work in cancer.

While certain genes highlighted during this analysis were already known to be associated with cancer, others were not – including genes present during the evolution of the earliest animals on Earth. Looking more closely into how these genes can cause disease may help us better understand and fight cancer.

DOI: https://doi.org/10.7554/eLife.40947.002

Through the addition of new genes and repurposing of existing genes, metazoan evolution has led to progressive formation of intricate and interconnected regulatory layers (*Arenas-Mena, 2017*; *Chen et al., 2014*; *Schmitz et al., 2016*), which suppress improper activation of replicative processes originating in single-celled organisms and ensure the complex phenotypes and cooperative growth required for multicellularity. Our investigation of gene expression data from a collection of solid tumors revealed extensive downregulation of genes specifying multicellular phenotypes and activation of genes conserved to unicellular organisms (*Trigos et al., 2017*). This was accompanied by significant loss of coordinated expression of unicellular and multicellular processes within tumors, suggesting selection for the disruption of key regulators mediating communication between unicellular and multicellular genes. Evidence for such selection comes from the clustering of cancer genes at the evolutionary boundary of unicellularity and multicellularity (*Domazet-Loso and Tautz, 2010*) and the accumulation of mutations in genes required for multicellular development during tumor progression (*Chen et al., 2015*).

Only a limited number of driver mutations are thought to be responsible for the transition from normal, healthy cell to a malignant state (*Martincorena et al., 2017*; *Miller, 1980*; *Schinzel and Hahn, 2008*; *Stratton et al., 2009*), suggesting individual mutations in highly connected genes in the regulatory network could bring about significant changes in cellular phenotypes. Under our model, the highest impact mutations would be those affecting proteins modulating the communication between the subnetworks that sustain multicellularity and the conserved core of fundamental cellular processes originated in single-celled organisms (*Trigos et al., 2018*). This would enhance a more primitive phenotype and provide a strong selective advantage for individual cellular lineages. Therefore, the contribution of somatic mutations to the rewiring of transcriptional networks during tumor development can be contextualized and estimated by their effect on gene regulatory networks pieced together during evolution, aiding the identification of key driver mutations.

Here, we elucidate how mutational heterogeneity across tumors results in common cellular hallmarks by accounting for their evolutionary ages and locations in the GRN. We found an overrepresentation of copy-number aberrations and point mutations in genes dating back to early metazoan ancestors. Point mutations disrupted key master regulators that evolved in early in the metazoan lineage, suggesting a primary role in the uncoupling of the subnetworks regulating multicellularity

and the fundamental core of cellular processes dating back to single-celled organisms. CNAs were involved in a complementary mechanism of dysregulation, generally disrupting the downstream targets of each of these subnetworks. These results indicate that both point mutations and CNAs contribute to the dysregulation of multicellularity in cancer, but do so in different ways, impacting regions of transcriptional networks with distinct roles in the regulation of multicellularity. Finally, we show how our approach of integration of sequence conservation and transcriptome data with annotated regulatory associations provides a framework for identifying important driver mutations and prioritizing compounds for targeted therapy, at the level of both patient populations and individual tumors.

## Results

### Early metazoan genes are enriched with point mutations and copy-number aberrations acquired during tumourigenesis

We investigated the association between the evolutionary ages of genes and the frequency of copy-number aberrations (CNAs) and point mutations across tumor cohorts (*The Cancer Genome Atlas Network, 2017a*; *The Cancer Genome Atlas Network, 2017b*). We collected CNA and point mutation data from 10256 and 9926 patients, respectively, from The Cancer Genome Atlas across 30 tumor types (see Materials and methods). We selected a subset of genes that were consistently amplified or deleted in at least 10% of patients of each tumor cohort, and genes with either missense or loss-of-function (LoF) mutations in at least three patients and with a higher rate of occurrence than synonymous mutations (*Figure 1—figure supplement 1*) (see Materials and methods). Human genes were classified by their evolutionary age using phylostratigraphy (*Domazet-Loso and Tautz, 2010*), resulting in 16 phylogenetic groups (phylostrata) (*Figure 1—figure supplement 2*), ranging from genes found in unicellular ancestors (Phylostrata 1–3) (6,719 UC genes), to genes found in early metazoans (Phylostrata 4–9) (7,939 EM genes), and mammal-specific genes (Phylostrata 10–16) (2,660 MM genes) (*Figure 1—figure supplement 3*) (*Trigos et al., 2017*).

We calculated the fraction of genes in each phylostratum with recurrent CNAs and point mutations, accounting for differences in CNA and mutation rates between tumor cohorts by ranking each phylostratum by the fraction of genes altered (*Figure 1A,B*). We found an increasing trend of enrichment of CNAs starting from the earliest UC genes (phylostratum 1), but peaking in EM genes (phylostrata 4–8), with EM genes being the most enriched with both amplifications and deletions across tumors. A high proportion of tumor types (14/30 tumor types for amplifications and 27/30 tumor types for deletions) had at least 3 EM phylostrata in the top five most recurrently altered phylostrata. In contrast, recurrent CNAs were consistently depleted from MM genes (phylostratum 10 onwards), indicating a lack selection for CNAs in younger genes. The decreasing enrichment trend along the phylostrata was significant for amplifications in 20/30 tumor types, and for deletions in 27/30 tumor types (Jonckheere-Terpstra tests Benjamini-Hochberg adjusted p<0.05). Similar results were obtained using recurrent gains/losses as defined by Gistic2, as the one used by Gistic2 (see Materials and methods, *Figure 1—figure supplement 4*), which employs a rigorous probabilistic model to identify recurrent CNAs. Among the recurrently amplified and deleted EM genes are well-known cancer genes. Examples include the EGFR oncogene, recurrently amplified in more than 10% of patients in 19/30 of the studied tumor types and having emerged together with bilaterians (Phylostratum 6), and the tumor suppressor TP53 which also dates back to early metazoan ancestors (Phylostratum 5) and is found recurrently deleted in an average of 12.73% of patients across all tumor types studied. In contrast to the patterns obtained for genes with recurrent CNAs, genes not recurrently copy-numbered altered (frequency <0.10 across patients) were not enriched in EM genes, but were often enriched in MM genes (*Figure 1—figure supplements 5* and *6*). An exception is TNFRSF17 (also known as BCMA, BCM), a mammal-specific gene involved in immune system processes that was amplified in >10% of BRCA, ACC, KIRP, THCA, PRAD and KIRC patients. TNFRSF17 has classically been associated with lymphomas (*Laâbi et al., 1992*) and has oncogenic properties (*Coquery and Erickson, 2012*; *Zhao et al., 2008*). Overall, our results suggest that EM genes are specifically preferentially under selection for recurrent CNAs across patients.

A similar enrichment of recurrent mutations in EM genes was identified for point mutations (*Supplementary file 1*). We found an increasing trend of enrichment of point mutations beginning

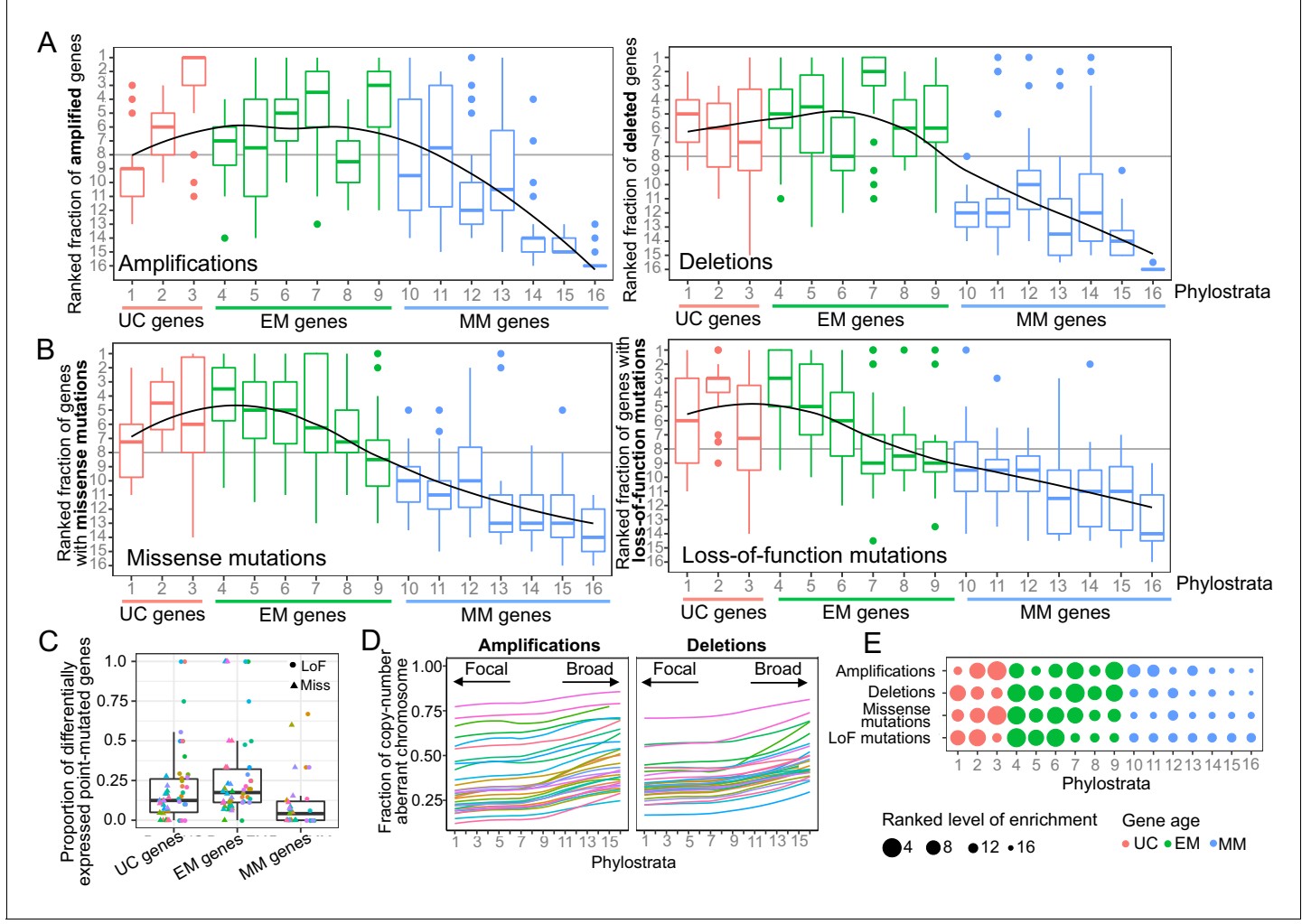

**Figure 1.** Enrichment of CNAs and point mutations in EM genes. (**A**) Fraction of amplified (left) and deleted (right) genes across phylostrata. EM genes are preferentially copy-number altered across tumor types, whereas MM genes are depleted. (**B**) Fraction of genes with missense (left) and LoF (right) mutations across phylostrata. Late UC genes and EM genes are enriched in missense and LoF mutations across tumor types, whereas MM genes consistently have the lowest fraction of genes with point mutations. The fractions rank from 1 (phylostratum with the highest fraction of mutated genes) to rank 16 (phylostratum with the lowest fraction of mutated genes). The line is a trendline calculated using a loess smoothing function. (**C**) Proportion of differentially expressed point-mutated EM genes across all tumor types compared to UC (one-sided Wilcoxon p=0.031) and MM genes (p=3.78×10$^{-6}$). (**D**) Presence of genes of each phylostratum at different fractions of chromosome altered by amplifications. Older genes are preferentially located in regions with focal alterations, whereas younger, MM genes are located in regions with broader changes, suggesting stronger selection for the CNA of UC and EM genes (increasing trend adj. p<0.05). (**E**) Summary enrichment results of recurrent point mutations and CNAs in phylostrata across tumors. The size of the point corresponds to the level of enrichment (rank). The largest enrichment occurs in EM genes, with some enrichment of UC genes.

DOI: https://doi.org/10.7554/eLife.40947.003

The following figure supplements are available for figure 1:

**Figure supplement 1.** Ratio of the number of missense and LoF mutations over synonymous mutations.
DOI: https://doi.org/10.7554/eLife.40947.004

**Figure supplement 2.** Phylogenetic tree depicting gene phylostrata.
DOI: https://doi.org/10.7554/eLife.40947.005

**Figure supplement 3.** Number of genes in each phylostratum.
DOI: https://doi.org/10.7554/eLife.40947.006

**Figure supplement 4.** Enrichment in EM genes of recurrent CNAs and point mutations identified by Gistic and MutSig2CV.
DOI: https://doi.org/10.7554/eLife.40947.007

**Figure supplement 5.** Fraction of non-recurrent amplified genes.
DOI: https://doi.org/10.7554/eLife.40947.008

*Figure 1 continued on next page*

*Figure 1 continued*

**Figure supplement 6.** Fraction of non-recurrent deleted genes.
DOI: https://doi.org/10.7554/eLife.40947.009
**Figure supplement 7.** Fraction of non-recurrent missense mutations.
DOI: https://doi.org/10.7554/eLife.40947.010
**Figure supplement 8.** Fraction of non-recurrent loss-of-function mutations.
DOI: https://doi.org/10.7554/eLife.40947.011
**Figure supplement 9.** Presence of genes by phylostratum by fraction of aberrant chromosome.
DOI: https://doi.org/10.7554/eLife.40947.012

at genes that date back to later unicellular ancestors (phylostratum 2), but peaking in early metazoan genes across tumors, with genes dating to the earliest metazoans (phylostratum 4) being the most enriched (*Figure 1B*, *Figure 1—figure supplement 4C*). In tumor types where $\geq 5$ phylostrata had recurrent missense (23 overall) or LoF mutations (14 overall), at least three of the five phylostrata most enriched for missense or LoF mutations were EM in 15/23 tumor types and 11/14 tumor types, respectively. In contrast, MM genes were consistently depleted of recurrent point mutations. The decreasing trend of enrichment associated with gene age was significant in the majority of tumor types for both missense (18/23) and LoF (12/14) mutations (Jonckheere-Terpstra tests Benjamini-Hochberg adjusted p<0.05). In contrast, this enrichment pattern was not observed for non-recurrently point-mutated genes (*Figure 1—figure supplements 7* and *8*).

To determine the effect of somatic mutations on transcriptional states, we calculated the proportion of genes in each of our major phylostratigraphic groups where differential expression could be tied to mutation status at the per-patient level (*Figure 1C*). EM genes with missense or LoF mutations were more likely to be differentially expressed when point mutated than were UC or MM genes (one-sided Wilcoxon tests p=0.031 and p=3.78×10$^{-6}$, respectively). These results suggest that the strong selection for point mutations in EM genes across tumors could be linked to their dysregulation of expression providing an advantage to tumor development.

We also investigated the association between gene age and signatures of selection for CNAs at the chromosome level (*Figure 1D*, *Figure 1—figure supplement 9*). For each patient and chromosome, we associated the evolutionary ages of the amplified or deleted genes with the fraction of chromosome affected by those CNAs, across tumor types (*Figure 1D*). There was a strong and significant increasing trend along the phylostrata (BH adjusted p<0.05) for amplifications in all tumor types and for deletions in >80% tumor types. These trends were not restricted to particular chromosomes (*Figure 1—figure supplement 9*). UC and EM genes were preferentially located in focally copy-numbered altered regions, suggesting stronger localized selection for the CNA of UC and EM genes. In contrast, MM genes were located in regions of broad copy-number changes, suggesting the CNA of MM genes are likely passenger events swept up in the large chromosomal rearrangements that occur during cancer development.

Our results indicate a preferential recurrent alteration by both CNAs and point mutations of EM genes across tumor types (*Figure 1E*, *Figure 1—figure supplement 4D*), suggesting that disruption of these genes by genetic changes likely provides an advantage in the development of multiple tumor types, whereas mutations in genes that evolved later in metazoan evolution, namely MM genes, are unlikely to be playing a significant role.

## Point mutations and CNAs acquired during tumor development differentially affect the human regulatory network

The observed enrichment patterns across cancer types suggested alteration of EM genes provides a selective advantage to tumors. Known cancer drivers are mostly of EM origin (*Domazet-Loso and Tautz, 2010*) and are highly interconnected in human molecular networks (*Cheng et al., 2014*), suggesting EM genes hold regulatory roles with important pleiotropic effects in cancer (*Trigos et al., 2018*). Since important innovations required for the regulation of transcriptional networks from unicellular ancestors evolved in early metazoan species, we investigated whether this could be evidenced in the current structure of the human gene regulatory network (GRN). The GRN was obtained by subsetting the network from PathwayCommons (*Cerami et al., 2011*; *Pathway Commons, 2017*) to include only edges annotated with control-of-expression.

Given the directed nature of the GRN, regulator genes can be distinguished from downstream target genes (*Figure 2A*). As expected, many more genes act as targets (12,812) than as regulators (1,370), indicating the presence of key regulatory hubs regulating a multitude of target genes. We found that over half (56.42%) of regulators in the GRN were EM genes, whereas only 37.88% and 5.69% were UC and MM genes, indicating an enrichment of EM genes as regulators (Fisher enrichment test p=$6.48 \times 10^{-6}$) (*Figure 2B*). Focusing on the genes with key regulatory roles, we investigated regulators with at least 10 downstream targets (out-degree >= 10), which correspond to the upper quantile of the distribution of out-degree across all regulators (*Figure 2—figure supplement 1*). We found that 65.12% of these master regulators were EM genes, whereas only 28.49% and 6.40% were UC and MM genes, respectively, indicating that key master regulators with the largest pleiotropic effects in the network were mostly EM genes. This structure of the GRN substantially differed from general protein-protein interaction (PPI) networks (e.g. (*Cerami et al., 2011*; *Chatr-Aryamontri et al., 2017*; *Li et al., 2017*)) where UC genes are usually the most connected (*Figure 2—figure supplement 2*), suggesting specific evolutionary processes shaping the GRN resulted in key regulatory roles for EM genes.

Additionally, EM genes were the most highly regulated downstream genes in the GRN, measured by the number of incoming edges (in-degree), with EM genes having an average of 8.76 incoming edges, compared to only 6.59 and 4.54 in UC and MM downstream target genes (Wilcoxon test p<$2.2 \times 10^{-16}$ in both cases) (*Figure 2C*, *Figure 2—figure supplement 3*). The enrichment of EM genes as both regulators and highly regulated downstream targets in the GRN indicates that gene regulation in humans is predominantly under control of EM genes (*Figure 2D*). Therefore, we hypothesized that the preference of somatic mutations in EM genes might stem from their key regulatory roles in the human GRN.

To test this, we assessed whether selection for somatic mutations in EM genes was linked to their central regulatory roles (*Figure 2A*). Given that broad CNAs involving large chromosome sections include a high number of genes with poor resolution of the genes under selection, we focused on recurrent CNAs that included less than 25% of the genes of a chromosome in at least 10% of patients of each tumor cohort. We found that a higher fraction of regulators were affected point mutations (mean fraction altered = 0.20) than CNAs (0.12) in 80.77% of tumor types with at least three recurrent point mutations or CNAs (Wilcoxon test p=$2.95 \times 10^{-5}$) (*Figure 2E*), with LoF mutations driving most of the signal (*Figure 2—figure supplement 4*). In contrast, downstream target genes without a regulatory role were more likely to be affected by CNAs (Wilcoxon test p=$2.95 \times 10^{-5}$) (mean fraction altered = 0.88 for CNAs, 0.80 for point mutations) (*Figure 2E*).

This dichotomy was even more pronounced in the top 5% most recurrently mutated genes, those with recurrent mutations in ≥7 tumor types. Over half (54%) the regulators in this set are recurrently point mutated, but only 13% are recurrently CNA, a fourfold difference. In contrast, 87% of the most highly mutated targets are affected by CNAs but only 46% are affected by point mutations. Similar results were obtained when we used the significantly CNA and point mutated genes identified by MutSig2CV and Gistic algorithms (*Figure 2—figure supplement 5*). Therefore, recurrent point mutations are more likely to affect master regulators in the GRN, whereas recurrent CNAs are more likely to affect target genes.

The complex regulatory interactions in the GRN result in many genes having a dual role, acting as both regulators and targets, with EM genes being both master regulators and under high degree of regulation (*Figure 2B–D*). To account for this dual role, we calculated the ratio of the number of outgoing edges (out-degree) to the number of incoming edges (in-degree), with greater ratios indicating a predominantly regulatory role (*Figure 2F*, *Supplementary file 2*), and calculated the median value across all genes with a dual role for each tumor type. We found that EM genes with point mutations held stronger regulatory role across tumors (median ratio = 0.83) than UC and MM genes with point mutations (median ratio = 0.51 and 0.31, respectively). In contrast, EM genes with CNAs were skewed toward being highly regulated downstream targets (median ratio = 0.38), even compared to UC and MM genes with CNAs (median ratio UC = 0.67 and MM = 0.60). Similar results were obtained using MutSig2CV and Gistic to define recurrent point mutations and CNAs (*Figure 2—figure supplement 6*). Therefore, the observation of selection for point mutations in EM regulators and CNAs in EM target genes also holds for genes with dual regulatory and target roles in the GRN.

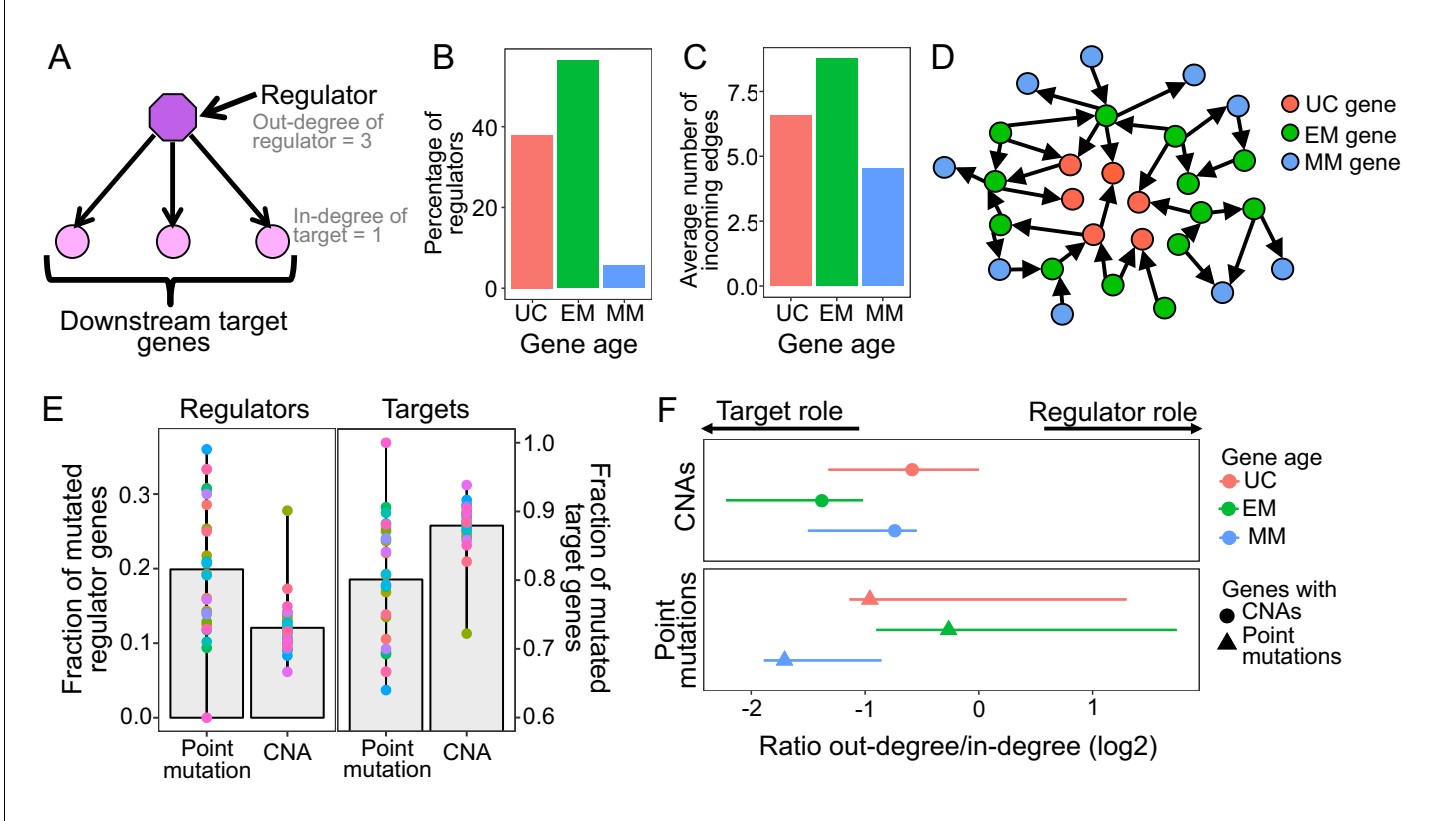

**Figure 2.** Point mutations in EM genes affect mostly regulators, whereas CNAs in EM genes affect downstream targets. (A) Diagram of a GRN distinguishing regulator and target genes. The number of outgoing edges from a regulator corresponds to its out-degree, whereas the number of incoming edges to a target gene is denoted by its in-degree. (B) Percentage of regulators of each age. Regulators are enriched in early metazoan genes (Fisher enrichment test p=$6.48\times10^{-6}$), with over half being EM (56.42%). (C) Average number of incoming edges for targets of each age. EM genes are also among the mostly highly regulated genes, with an average of 8.76 regulators controlling their activity, compared to 6.59 and 4.54 regulating UC and MM downstream target genes. (D) GRN diagram. EM genes (green) are highly interconnected, acting as master regulators and highly regulated targets. (E) Fraction of mutated regulator and target genes by each mutation type. A greater proportion of regulators are affected by recurrent point mutations than CNAs (0.20 vs 0.12; left), whereas the opposite trend is observed for targets (Wilcoxon test p=$2.95\times10^{-5}$). (F) Ratio of out-degree/in-degree (log2) of genes with mutations. EM genes are strongly biased toward a preferential regulatory role when point mutated, whereas the CNAs of EM genes preferentially occurs in those with a strong downstream target role. Points represent the median values for each tumor type and bars represent the range between the upper and lower quantiles.

DOI: https://doi.org/10.7554/eLife.40947.013

The following figure supplements are available for figure 2:

**Figure supplement 1.** Distribution of out-degree of regulators in the GRN.
DOI: https://doi.org/10.7554/eLife.40947.014

**Figure supplement 2.** Degree of genes across multiple molecular networks.
DOI: https://doi.org/10.7554/eLife.40947.015

**Figure supplement 3.** Distribution of in-degree of target genes in the GRN.
DOI: https://doi.org/10.7554/eLife.40947.016

**Figure supplement 4.** Fraction of regulators and targets with mutations.
DOI: https://doi.org/10.7554/eLife.40947.017

**Figure supplement 5.** Fraction of mutated regulator and target genes by each mutation type.
DOI: https://doi.org/10.7554/eLife.40947.018

**Figure supplement 6.** Ratio of out-degree/in-degree (log2) of genes with mutations.
DOI: https://doi.org/10.7554/eLife.40947.019

Although there is a recurrent selection across tumor cohorts for the somatic mutation of EM genes, our results reveal that selection for point mutations and CNAs differentially disrupt the GRN. EM hub genes with key regulatory roles were preferentially disrupted by point mutations, indicating that few point mutations in key regulators are more likely to create large disruptions across the GRN. In contrast, downstream target genes were preferentially affected by recurrent CNAs, and are therefore more likely to have a localized effect.

## Point mutations disrupt the regulation between UC and EM genes

A main characteristic of cancer development is the loss of coordination in expression between UC and MC genes together with overexpression of UC genes and downregulation of MC genes (*Trigos et al., 2017*), suggesting a compartmentalization of the GRN into UC and MC gene network regions interconnected by key regulatory links that get disrupted by mutations during cancer development (*Trigos et al., 2018*) (*Figure 3A*).

To distinguish these UC and EM network regions, we calculated the percentage of downstream UC and EM target genes for each regulator, and classified individual regulators as preferentially regulating UC targets (>2/3 UC genes) (UC-t regulators), EM targets (>2/3 EM genes) (EM-t regulators), MM targets (>2/3 MM genes) or being at the interface of UC and EM targets by regulating a mix of UC and EM downstream targets (>1/10 UC and EM genes) (UC/EM-i regulators) (*Figure 3B*, main panel, *Supplementary file 3*). We excluded regulators that primarily controlled mammalian genes from further analysis, as they only accounted for 2.04% of all regulators. Regulators not meeting any of the above criteria were also excluded (10.36%). UC-t regulators mostly dated back to UC ancestors (50.81%, Fisher test p=0.021) while both EM-t regulators (56.68%, Fisher test p=0.0011) and UC/EM-i regulators (61.73%, Fisher test p=0.00022) were mostly comprised of EM genes.

We next investigated how point mutations and CNAs differentially affected UC-t, EM-t and UC/EM-i regulators. Given dysregulation of UC and MC gene expression and co-regulation has been found to be consistent across multiple tumors types (*Trigos et al., 2017*) and therefore likely to share similar drivers, we only considered the top 5% most recurrently mutated genes across tumor cohorts. The majority of regulators with recurrent point mutations were UC/EM-i (85.71%), whereas this percentage was only 32.81% and 32.33% for regulators affected by CNAs or not recurrently mutated (*Figure 3B* density plot, *Figure 3C*, *Figure 3—figure supplement 1*), indicating preference for the point mutation of UC/EM-i regulators, with no preference for those affected by CNA. These results indicate overrepresentation of highly recurrent point mutations across tumor cohorts in regulators at the UC/EM interface (*Figure 3—figure supplement 2*), which was not observed for CNA regulators or non-recurrently mutated regulators.

To examine the functional downstream effects of point mutations in regulators, we calculated the percentage of downstream targets that were differentially expressed after point mutations in their regulator genes, and classified the magnitude of the downstream effect as being of low impact if less than 5% of the downstream genes were affected, and high impact otherwise. The majority of high impact mutations occurred in UC/EM-i regulators (75%), whereas only 43.13% of low impact mutations occurred in these regulators, indicating that high-impact mutations preferentially occur in in UC/EM-i regulators (two-proportions Z-test p=0.026) (*Figure 3D*). The actual ages of the UC/EM-i regulators were also associated with mutational impact. EM genes represented a higher proportion of UC/EM-i regulators with high impact mutations (67%) than those with low impact mutations (50%), and as a whole, point-mutated early metazoan regulators were two-times more likely than other point-mutated regulators to have a high impact (*Figure 3—figure supplement 3*). Thus, point mutations creating the most substantial alterations to gene expression tend to be in regulator genes of early metazoan origin at the interface of UC and EM genes.

Overall, our results suggest that the recurrence across tumor cohorts of point mutations in EM regulators is tied with transcriptional disturbances of the regulation between UC and EM genes in the GRN, making them potential gene drivers. Multiple known cancer genes were found among UC/EM-i regulators, including RB1, PIK3CA, SMAD4, NF1 TP53, TERT and MDM4. Functional enrichment analysis revealed that these regulators are involved in multiple signaling pathways, such as receptor tyrosine kinases, MAPK and PI3K-Akt signaling, but they are also involved in a diversity of other processes such as mitochondrial biogenesis and lipid metabolism (*Supplementary file 4*). Investigation of the pathways associated with UC/EM-i regulators revealed that their point mutation often had a significant effect on pathway expression levels (*Supplementary file 5*). Point mutation of

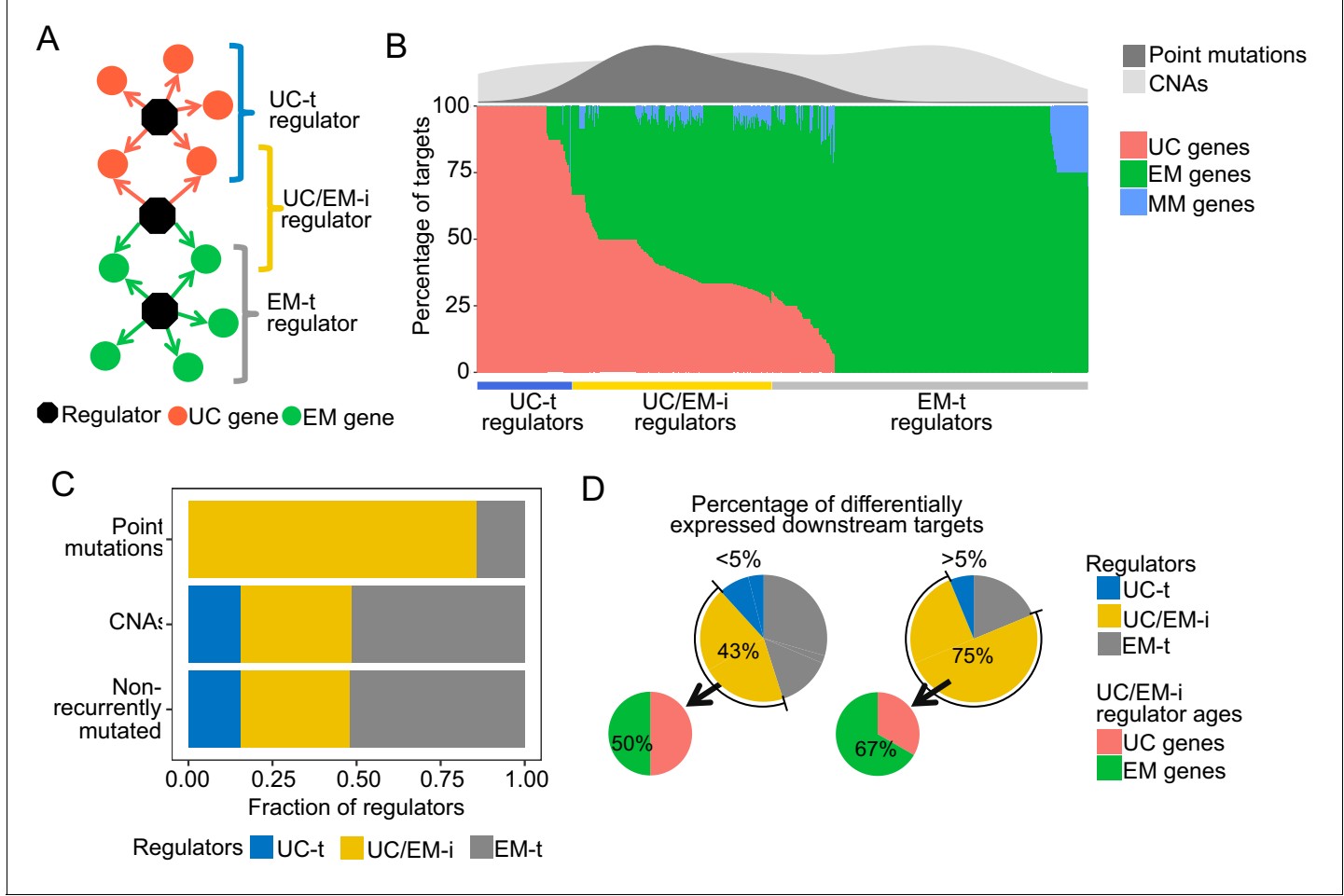

**Figure 3.** Point mutations in regulators affect UC-EM gene regulation. (**A**) Classification of regulators by the age of their downstream targets. UC-t regulators mostly regulate UC genes, EM-t regulators EM genes, and UC/EM-i regulators are at the interface of UC and EM genes. (**B**) (Lower panel) Percentage of UC, EM and MM target genes in regulators. (Upper panel) Distribution of recurrent point mutations (dark grey) and CNAs (light grey) across regulators. UC/EM-i regulators are enriched in point mutations. (**C**) Fraction of regulators with point mutations, CNAs and those non-recurrently altered. More than 85% of regulators affected by point mutations are UC/EM-i regulators. The fraction of regulators of each class affected by CNAs is similar to those not affected by recurrent mutations, indicating a lack of preferential alteration of a particular regulator class by CNAs. (**D**) Effect of point mutations in regulators on the expression of downstream targets. Point mutations with a high downstream effect (>5% differentially expressed targets) are more likely to be UC/EM-i regulators of EM origin. Low impact mutations (<5% differentially expressed targets) affect a higher proportion of regulators of a UC origin.

DOI: https://doi.org/10.7554/eLife.40947.020

The following figure supplements are available for figure 3:

**Figure supplement 1.** Distribution of recurrent point mutations and CNAs identified by MutSig2CV and Gistic in regulators.
DOI: https://doi.org/10.7554/eLife.40947.021
**Figure supplement 2.** Distribution of CNAs and point mutations across recurrently mutated regulators.
DOI: https://doi.org/10.7554/eLife.40947.022
**Figure supplement 3.** Effect of point mutations in regulators on the expression of downstream genes.
DOI: https://doi.org/10.7554/eLife.40947.023

well-studied genes such as p53 affected the expression of pathways such as MAPK signaling, p53 signaling, WNT, apoptosis, cell cycle across 18 tumor types; mutations in PTEN with changes in expression of p53 signaling in six tumor types, PIK3CA affecting the expression of multiple pathways such as ERBB signaling, MTOR, JAK-STAT, and VEGF signaling across 10 cancer types; and point mutations in SMAD4 in colon cancer affected WNT signaling. We also found other understudied effects. For example, mutation of EP300 was associated with differences in expression of cell cycle

and JAK-STAT pathways in bladder, endometrial, cervical and endocervical cancers. Point mutation of PIK3R1 affected ERBB, MTOR and JAK-STAT signaling. Intriguingly, our analysis pointed to other other EM UC/EM-i regulator genes with mutational frequencies and percentage of differentially expressed downstream targets similar to known cancer genes. Many of these genes, including KEAP1, HNF1A, NFE2L2 and LRRK2, have implied roles in cancer but no strong mechanistic links to date (Figure 5; Discussion).

## CNAs directly regulate the expression of UC and EM downstream targets in the GRN

While somatic mutation of regulators could provide a major selective advantage via simultaneous dysregulation of a multitude of downstream target genes, where and when such mutations occur is largely based on stochastic events during tumor development. An alternate and complementary mechanism for disrupting conserved regions of the GRN without mutation of master regulators would be direct mutation of downstream target genes, as suggested by our finding that CNAs predominantly affected target genes of the GRN (*Figure 2E–F*).

To investigate the contribution of CNAs to the disruption of the regulatory links between UC and EM genes, we calculated the fraction of downstream targets with CNAs for each regulator in each individual patient. To exclude possible redundant mechanisms resulting from CNAs in both regulators and targets, we only included in the analysis samples where the regulator was copy-number normal (CNN). We found that only a small fraction of downstream target genes of UC/EM-i regulators were CNA (median = 0.11), whereas a significantly larger fraction of downstream target genes of UC-t and EM-t regulators were affected (median = 0.25 and 0.33, respectively) (p=$3.91 \times 10^{-8}$ and p=$8.42 \times 10^{-27}$ comparing the fraction of CNA targets of UC/EM-i with that of UC-t and EM-t regulators, respectively) (*Figure 4A*). This indicates that CNAs preferentially affect target genes of UC-t and EM-t regulators, rather than directly disrupting the regulatory links at the interface of UC and EM regions of the GRN.

We hypothesized the preferential of CNAs in targets genes of UC-t and EM-t regulators would be associated with the direct transcriptional modulation of genes by CNAs. To test this, we calculated the expression fold-change in tumor samples with respect to their paired normal samples, and used Wilcoxon tests to compare fold-change values in samples where the gene was CNA and those where the gene was CNN. We found a higher percentage of target genes than regulators were differentially expressed after amplifications across all tumor types, and for deletions in 64.64% of tumors types, indicating that CNAs more strongly influence the expression of target genes than regulator genes (*Figure 4—figure supplement 1*). Specifically, UC target genes showed the largest changes in expression after CNAs (median values across tumor types: 7.80% upregulated after amplifications, 7.48% downregulated after deletions), followed by EM genes (4.45% and 4.79%, respectively), and lastly MM genes (2.46% and 1.62%) (*Figure 4B*, *Figure 4—figure supplement 2*) (Jonckheere-Terpstra decreasing trend test: amplifications p-value: 0.0028, deletions p-value: 0.0021), indicating that UC and EM targets are more susceptible to changes in expression after CNA.

However, the effect of CNAs on the expression of targets was dependent on their regulator. Targets of UC-t regulators were more likely to be differentially expressed after amplifications or deletions were the targets of UC/EM-i regulators (*Figure 4C*). On the other hand, target genes of EM-t regulators mostly modulated their expression in response to deletions as opposed to amplifications (one-sided Wilcoxon test p=0.043) (*Figure 4C*), suggesting selection of CNAs in targets of EM-t regulators could be a mechanism for the direct down regulation of EM network regions, as an alternative to direct mutation of their regulators. In contrast, CNAs of the target genes of UC/EM-i regulators changed their expression much less often, no matter whether it was amplified or deleted (two-sided Wilcoxon test p=0.95), suggesting CNAs are playing a less prominent role in the dysregulation of UC and EM interface regions than do somatic point mutations.

A model of transcriptional changes in UC-t and EM-t targets driven by CNAs as an alternative to direct mutation of the regulators themselves would predict the mutual exclusivity of concurrent CNAs of a regulator and its targets in the same tumor, as the co-occurrence of such mutations would be largely redundant. To test this hypothesis, we calculated the median fraction of targets with CNAs for each regulator across all patients in the tumor cohorts, and found that the fraction of targets with CNAs was significantly higher in patients where the regulator was CNN than when the

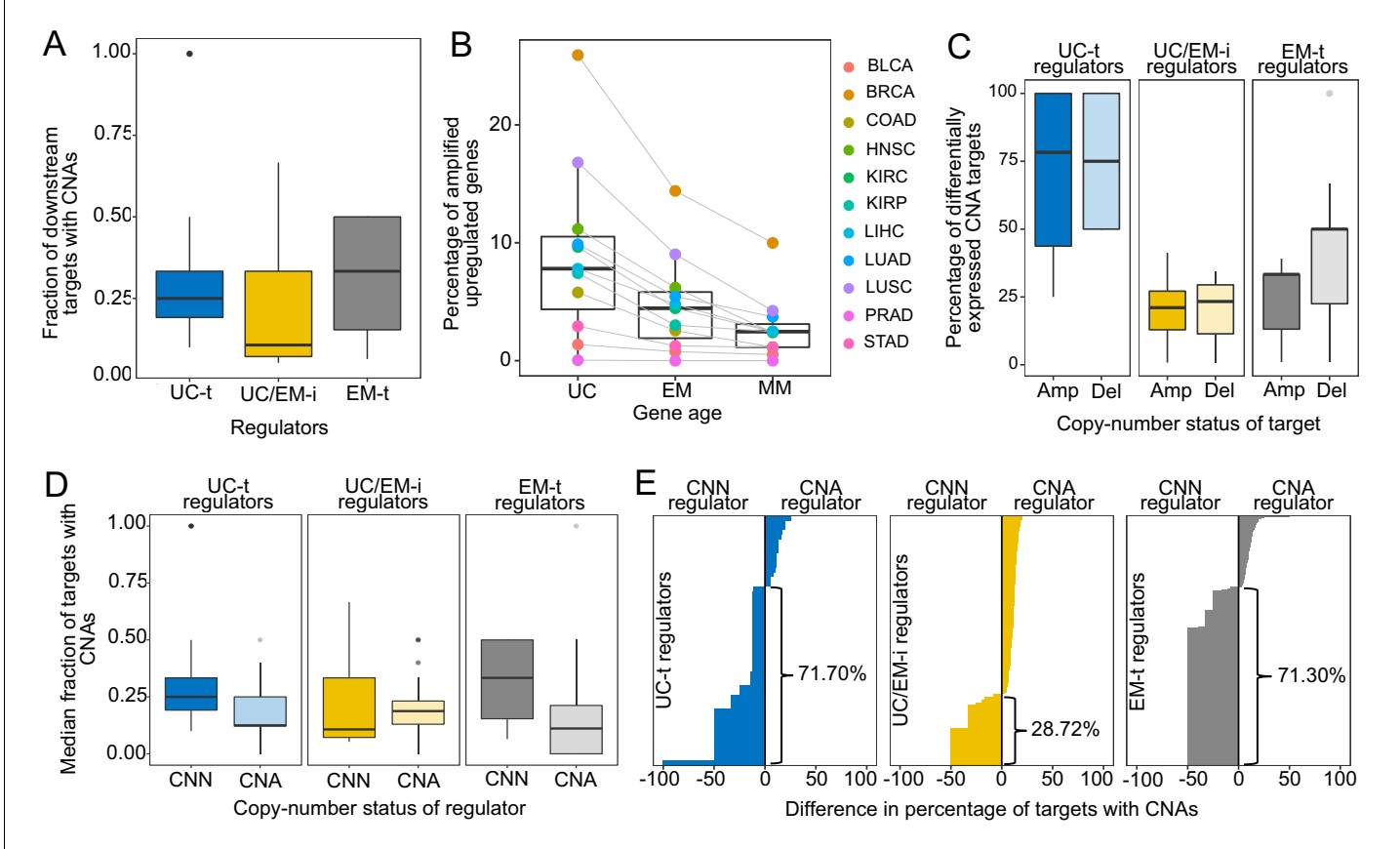

**Figure 4.** CNAs directly regulate the expression of UC and EM target genes. (**A**) Fraction of downstream targets with CNAs in regulators. Targets of UC-t and EM-t regulators are more likely to be affected by CNAs than targets of UC/EM-i regulators. (**B**) Percentage of differentially expressed target genes with amplifications. UC and EM target genes are more likely to be upregulated after amplifications compared to younger, mammal-specific genes (Jonckheere-Terpstra decreasing trend test p-value: 0.0028). A similar trend is found for the downregulation of deleted genes (*Figure 4—figure supplement 2*). (**C**) Median percentage of differentially expressed CNA genes per regulator class across tumors. Amplified and deleted target genes of UC-t regulators are more likely to be differentially expressed (median 78.27% and 75.00%, respectively), whereas few CNA target genes of UC/EM-i regulators are DE (median 21.05% and 23.33%, respectively). Deleted targets of EM-t regulators are more likely to be downregulated (50.00%) than amplifications are upregulated (33.33%). No preference is evident for targets of UC/EM-i regulators. (**D**) Fraction of target genes with CNAs when their regulators are CNA or CNN. A higher fraction of target genes are CNA when UC-t and EM-t regulators are CNN than when they are CNA (Wilcoxon test p=$1.36\times10^{-7}$ and p=$1.11\times10^{-42}$, respectively), indicating a preference for the alteration of targets of these regulators. However, UC/EM-i regulators display the opposite trend, although not significant. (**E**) Difference in the fraction of downstream targets altered by CNAs when their regulators are CNN or CNA. Values less than 0 indicate a higher fraction of CNA targets when the regulator is CNN. This trend is evident across UC-t regulators (71.70%) and EM-t regulators (71.30%), but not for UC/EM-i regulators (28.72%).

DOI: https://doi.org/10.7554/eLife.40947.024

The following figure supplements are available for figure 4:

**Figure supplement 1.** Targets are more likely differentially expressed after CNAs than regulators.

DOI: https://doi.org/10.7554/eLife.40947.025

**Figure supplement 2.** Percentage of differentially expressed target genes with CNAs.

DOI: https://doi.org/10.7554/eLife.40947.026

**Figure supplement 3.** The median fraction of targets with CNAs by regulator status.

DOI: https://doi.org/10.7554/eLife.40947.027

**Figure supplement 4.** Difference in the fraction of downstream targets altered by CNAs when their regulators are CNN or CNA.

DOI: https://doi.org/10.7554/eLife.40947.028

regulator was CNA (Wilcoxon test p-value=$4.10\times10^{-29}$, *Figure 4—figure supplement 3*). However, this trend was only observed for targets of UC-t and EM-t regulators (one-sided Wilcoxon test p=$1.36\times10^{-7}$ and p=$1.11\times10^{-42}$, respectively), but not for targets UC/EM-i regulators (p=0.79), where the trend seemed to be the opposite (*Figure 4D*), indicating that preferential alteration of

target genes by CNAs, as opposed to regulators, is a mechanism employed for the independent modulation of UC and EM network regions.

We next investigated how the copy-number status of regulator (CNA or CNN) was associated with the fraction of CNA downstream target genes (*Figure 4E*). We found that over two thirds of UC-t and EM-t regulators (71.70% and 71.30%, respectively) had a higher fraction of copy-aberrant downstream genes when they were CNN than when they were CNA. These results were independent of the evolutionary ages of the regulators, since the percentages were similar for both UC and EM regulators (*Figure 4—figure supplement 4*). A potential explanation for this pattern is illustrated by MDM2, where 33.33% of its downstream targets were CNA when MDM2 was CNN, but no target was CNA when MDM2 had changed in copy-number. Since MDM2 is an inhibitor of p53, either amplification of MDM2 or deletion of p53 would have a similar effect, and therefore there is no selection for the simultaneous co-occurrence of both alterations in the same patient. In contrast to the strong trends observed for UC-t and EM-t regulators, this trend was only observed in 28.72% of UC/EM-i regulators. The preferential copy-number aberration of targets of CNN UC-t and EM-t regulators, but not in CNN UC/EM-i regulators, supports our proposed model of selection for the CNA of target genes of UC-t and EM-t regulators, but not for those of UC/EM-i regulators.

Here we found that CNAs are a widespread mechanism of dysregulation of UC and EM target genes, directly modulating the expression of targets of UC-t and EM-t regulators. The effect of CNAs on the GRN is therefore distinct to the disruption by point mutations of the regulatory links between UC and EM genes, but are also important drivers of large transcriptional disturbances in tumors.

## UC/EM-i regulators are important drivers of tumourigenesis and influence drug sensitivity

UC/EM-i were found to be preferentially targeted by point mutations and are likely to be key points of vulnerability to cancer development given their regulatory role modulating UC and EM genes (*Trigos et al., 2018*). Therefore, we investigated these regulators as potential drivers of tumourigenesis and their role in determining drug response.

We compiled a set of known cancer drivers from the Cancer Census COSMIC database (*Forbes et al., 2017*). These genes were enriched in EM genes (Fisher test p=0.0015, 57.36% EM genes, 39.58% UC genes and 3.06% MM genes). 36.71% of the known cancer genes were regulators and were enriched in UC/EM-i regulators (46.88%, Fisher test p=0.0043), but depleted in UC-t (p=0.83) and EM-t regulators (p=0.95) (*Figure 5A*), supporting modulation of regulation between UC and EM genes is a common effect of cancer drivers. Furthermore, investigating drivers identified from multi-region sequencing data (*Caravagna et al., 2018*) in individual patients of non-small-cell lung cancers (*Jamal-Hanjani et al., 2017*) and breast cancers (*Yates et al., 2015*) revealed that 90.91% of lung cancer patients and 70.83% of breast cancer patients had at least one UC/EM-t regulator identified as a clonally mutated driver (*Figure 5—figure supplement 1*), supporting a fundamental role of the mutation of these regulators in cancer development.

We further investigated the importance of these regulators to cancer development, we used the dependency scores from CRISPR-Cas9 essentiality screens of 364 solid-tumor tissue cell lines from the Avana CRISPR-Cas9 genome-scale knockout dataset made available by Project Achilles and the Cancer Dependency Map project (*Meyers et al., 2017*; *Broad Institute, 2018b*). A high probability of dependency indicates a gene is essential for proliferation of a given cell line (*Figure 5—figure supplement 2*) (*for further details, see Meyers et al., 2017*). We calculated the odds ratio (OR) of each regulator type having a large effect on cell line proliferation (probability of dependency >= 0.95), with OR greater than one indicating increased likelihood of a high dependency (*Figure 5B*, *Figure 5—figure supplement 3*). As expected, we found most cell lines were highly dependent on UC-t regulators (OR >1 in 96.98% of cell lines), likely due to their role in regulating fundamental functions required for cell survival. However, we also found that 92.86% cancer cell lines were highly dependent on UC/EM-i regulators (OR consistently greater than 1), indicating these regulators are fundamental for cancer cell survival. In contrast, cancer cell lines were not nearly as dependent on EM-t regulators for their proliferation (OR >1 in only 0.55% of cell lines), indicating dysregulation of EM processes might contribute to tumourigenesis, but not be sufficient by themselves. These results suggest that UC/EM-i regulators are indispensible for cancer cell line proliferation.

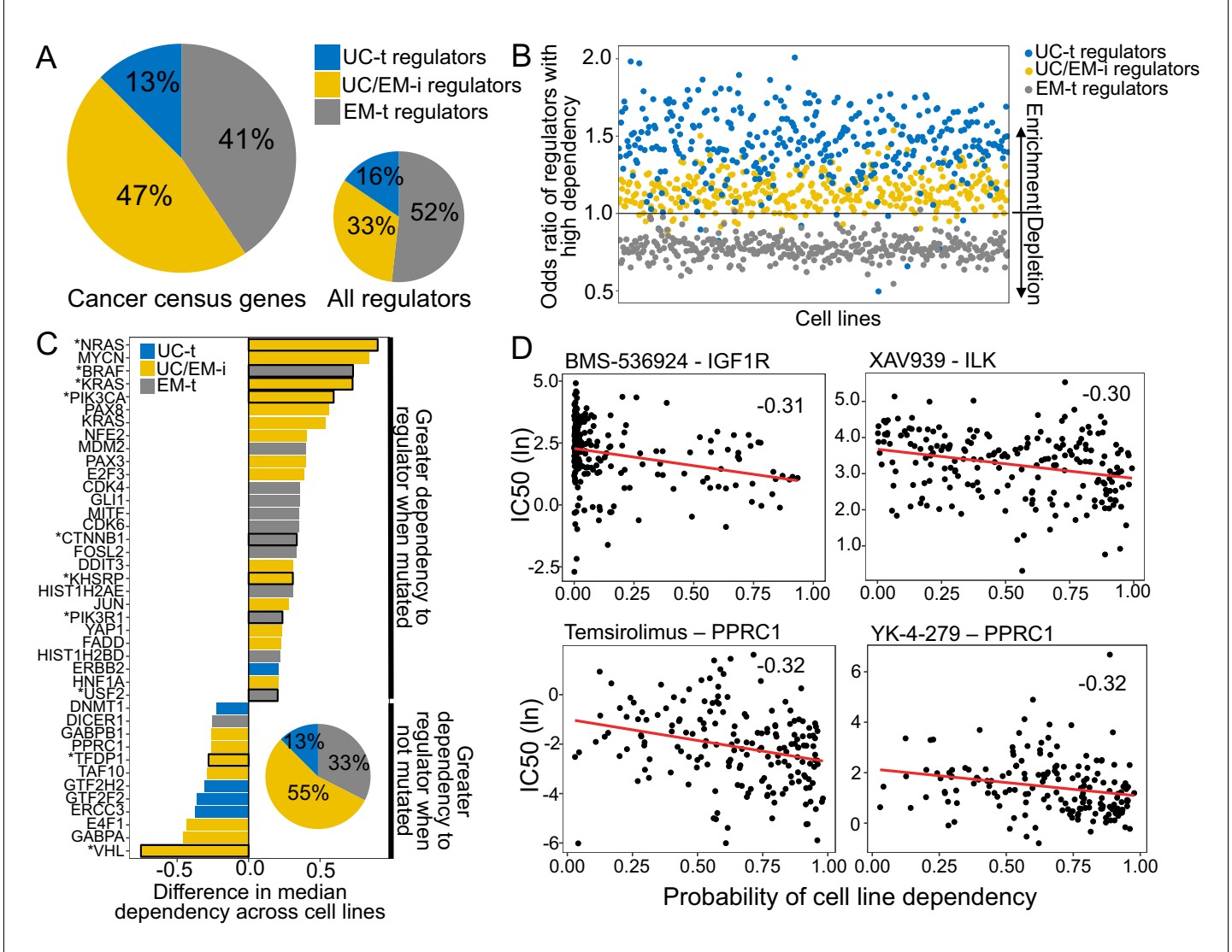

**Figure 5.** UC/EM-i regulators are fundamental to tumor development and drug response. (A) Fraction of known cancer drivers of each regulator class. While only 33% of regulators are UC/EM-i, 47% of cancer drivers are UC/EM-i regulators, indicating an enrichment of this regulator class in genes involved in cancer development. (B) Enrichment of regulators to which cancer cell lines are dependent, as demonstrated by gene knockout. Dependency of cancer cell lines to regulators is associated with regulator class, with an enrichment of UC-t and UC/EM-i regulators and a depletion of EM-t regulators. (C) Difference in cell line regulator dependency associated with mutational status. Most cells increase their dependency to specific regulators with point mutations or amplifications, indicating that the mutation of these genes are important for cancer cell survival. This is especially true for UC/EM-i regulators (55%, pie chart). Only regulators with a difference in the median dependency of at least 0.2 are shown. Asterisks denote regulators with significantly different dependency scores between cell lines where the gene was point mutated and non-point-mutated, and the rest those that were significantly different between cell lines where the gene was amplified and non-amplified. (D) Correlation between the probability of cell line dependency to UC/EM-i regulators and the IC50 of drugs. (Top left) Expected association between the dependency to the IGF1R regulator and the sensitivity to the IGF1R- inhibitor, BMS-536924. (Top right) Cell lines dependent to ILK show a greater sensitivity to the B-catenin pathway inhibitor (XAV939). (Bottom row) Cell lines dependent on PPRC1 showed increased sensitivity to mTOR inhibitors (temsirolimus) and RNA helicase A inhibitors (YK-4–279).

DOI: https://doi.org/10.7554/eLife.40947.029

The following figure supplements are available for figure 5:

**Figure supplement 1.** Number of driver and clonal point mutated regulators in 100 non-small-cell lung cancers from the TRACERx study (*Jamal-Hanjani et al., 2017*) and 50 breast cancers (*Yates et al., 2015*) obtained by *Caravagna et al. (2018)*.
DOI: https://doi.org/10.7554/eLife.40947.030

**Figure supplement 2.** Distribution of probabilities of dependency to regulators from the Avana CRISPR-Cas9 genome-scale knockout dataset.
DOI: https://doi.org/10.7554/eLife.40947.031

*Figure 5 continued on next page*

*Figure 5 continued*

**Figure supplement 3.** Odds ratio of dependency of regulators in cancer cell lines after knockout at different probability cutoffs.
DOI: https://doi.org/10.7554/eLife.40947.032
**Figure supplement 4.** Correlation between cell line dependency and IC50 Distribution of correlations between cell line dependency to regulators and the IC50 (ln) of 250 drugs.
DOI: https://doi.org/10.7554/eLife.40947.033
**Figure supplement 5.** Significant correlations between the probability of dependency to UC/EM-i regulators and the IC50 (ln) of drugs.
DOI: https://doi.org/10.7554/eLife.40947.034
**Figure supplement 6.** Distribution of correlations of the probability of cell-line dependency to regulators and the IC50 (ln) of drugs.
DOI: https://doi.org/10.7554/eLife.40947.035
**Figure supplement 7.** Correlation between drug sensitivity per cell-life tissue type and PPRC1 cell-line dependency.
DOI: https://doi.org/10.7554/eLife.40947.036

We hypothesized the degree of dependency on UC/EM-i regulators would be tied to their mutational and copy-number status. For each regulator, we classified cell lines into those with or without a point mutation or amplification in the regulator based on data from the Cancer Cell Line Encyclopaedia (CCLE) (https://portals.broadinstitute.org/ccle) (*Broad Institute, 2013*; *Broad Institute, 2018a*), and calculated the median dependency scores for mutated and non-mutated cell lines (*Supplementary file 6*). We only considered regulators whose difference in median dependency between mutated and non-mutated cell lines was at least 0.20, and were significant by Wilcoxon tests ($p<0.05$). This revealed 40 regulators whose mutation was associated with cell-line dependency (*Figure 5C*). Multiple well-known cancer genes were among these top hits. Amplification of the oncogenes *ERBB2*, *CDK6*, *MDM2* affected dependency, as did point mutations in *PIK3CA*, *KRAS*, *NRAS*, *VHL* and *BRAF*, validating our approach to highlight genes with significance in cancer. Over half of these top hits correspond to UC/EM-i regulators (55.00%), indicating that mutations in UC/EM-i regulators are more likely to affect the dependency of a cancer cell line to a regulator. We found that in most cases cell lines were more dependent on a regulator when it was point mutated (80.00%) or amplified (66.67%) than when it was non-mutated, suggesting that mutation of these regulators are key to cancer cell line proliferation (*Figure 5C*).

We also investigated the implications of dependency to UC/EM-i regulators on drug sensitivity. Based on the half maximal inhibitory concentration values (IC50) from drug sensitivity screens covering 250 drugs from the Genomics of Drug Sensitivity in Cancer database (*Genomics of Drug Sensitivity in Cancer Consortium, 2016*; *Yang et al., 2013*), we calculated the Spearman correlation between dependency scores and the IC50 values (*Figure 5—figure supplement 4*), and identified 11 significant associations with UC/EM-i regulators, 38 with UC-t regulators and 23 with EM-t regulators (correlation $<-0.25$ and adj. $p<0.05$). These identified regulators were among the most highly correlated with the IC50 of the particular drug (*Figure 5—figure supplement 5*). As expected, we found strong correlation between dependency scores for UC/EM-i regulators and the IC50 of drugs targeting the regulator (e.g. IGF1R and Linsitinib, BMS-536924 and BMS-754807), as well as between *MAPK1* and an inhibitor of related genes in the MAPK/ERK pathway ((5Z)−7-Oxozeaenol), validating our approach (*Figure 5D*, *Figure 5—figure supplement 6*).

However, we also found unexpected strong correlations between the IC50 of particular drugs and the dependency scores of UC/EM-i regulators (*Figure 5D*, *Figure 5—figure supplement 6*). For example, the IC50 of XAV939, an inhibitor of Wnt/β-catenin, was also strongly correlated with the dependency to ILK (−0.30), a regulator of integrin-mediated signal transduction involved in tumor growth and metastasis, supporting the use of Wnt/β-catenin inhibitors for cancers dependent on ILK, including colon, gastric and ovarian and breast cancers (*Hannigan et al., 2005*). We also found strong correlation across cell lines between the dependency to PPRC1 and mTOR-inhibitors (temsirolimus, used in the treatment of renal cancer), dual PI3K/mTOR-inhibitors (dactolisib, in clinical trial for advanced solid tumors (*Wise-Draper et al., 2017*)), YK-4–279 (showing pre-clinical efficacy for Ewing sarcoma (*Lamhamedi-Cherradi et al., 2015*)) and the chemotherapy agent docetaxel, currently used in the treatment of breast, lung cancer, stomach cancer, head and neck and prostate cancer. Of the tumor types included in our study, the correlation with PPRC1 dependency was particularly strong ($<-0.25$) in liver, lung and stomach cell lines for temsirolimus sensitivity, lung and stomach cell lines for docetaxel and dactolisib sensitivity and breast cell lines for YK-4–279

sensitivity, but were also held for a number of other solid tumor types (*Figure 5—figure supplement 7*), suggesting their use across multiple cancer types. With this, our novel approach has identified understudied potential vulnerabilities for cancer development and proposed drug repositioning possibilities.

## Discussion

Detailed analyses of recurrent somatic mutations across tumor types revealed the prevalence of mutations related to both gene age and its position within the regulatory network. We provide evidence that point mutations and CNAs play complementary roles in the transcriptional dysregulation in cancer by affecting distinct regions of the underlying gene regulatory network, supporting the loss of communication between the core biological processes originating in ancient single-celled life and the regulatory controls acquired during metazoan evolution to control these processes. This would result in tumor convergence to similar transcriptional states of consistent activation of genes from unicellular ancestors and loss of cellular functions characteristic of multicellular organisms. Our results attribute key roles to genes at the interface of unicellular and multicellular regulation in tumourigenesis, with implications for conventional and experimental therapies.

Common hallmarks shared by tumors of diverse genetic backgrounds suggest the consequences of mutations acquired during tumor development follow common principles, promoting the downregulation of genes and pathways associated with multicellularity and the activation of fundamental cellular processes evolved in early unicellular organisms (*Trigos et al., 2017*). Here, we found genes central to the human gene regulatory network that arose in early metazoans were the most often recurrently affected by point mutations and CNAs across tumor types. Other studies have found that gatekeeper cancer drivers (those that regulate cell cooperation and tissue integrity) emerged at a similar evolutionary time, whereas caretaker genes (those ensuring genome stability) emerged at the onset of unicellular life (*Domazet-Loso and Tautz, 2010*). Our results suggest recurrent mutations mostly affect gatekeeper genes regulating fundamental aspects of multicellularity, whereas the disruption of caretaker activities by recurrent somatic mutations and CNAs is more limited.

We found the impact of point mutations and copy-number aberrations was concentrated on specific regions of the gene regulatory network. Point mutations preferentially affected gene regulators at the interface of unicellular and early metazoan subnetworks, likely affecting the regulation of multicellular components over fundamental cellular processes. We found that these interface genes were involved in well-known cancer signaling networks (e.g. TP53, NF1, PIK3CA), but found that other processes of emerging interest, such as metabolism and mithochondrial dysregulation are also associated with the rupturing of regulation of unicellular and multicellular components by point mutations. On the other hand, CNAs preferentially affected their downstream target genes, directly promoting the independent activation and inactivation of regions predominantly composed of unicellular and multicellular genes, as opposed to mixed regions, further supporting a loss of multicellularity and the tight association between gene expression level and gene age (*Trigos et al., 2017*).

Samples where UC/EM-i regulators were mutated tended not to have CNAs in the corresponding target genes and vice versa; in samples carrying CNAs for multiple target genes, unicellular and multicellular, the cognate UC/EM-i regulator were mostly unaltered. These complementary but distinct mechanisms of alteration to the same regulatory subnetworks by different classes of somatic mutations in different tumors provide mutually exclusive but highly convergent paths toward common hallmarks associated with the loss of multicellularity features in cancer.

Our evolutionary network analysis approach also highlights potential early driver genes by elucidating the evolutionary regulatory context in which genes operate. Only a handful of driver mutations are thought to be responsible for the transition from a normal, healthy cell to a malignant state (*Martincorena et al., 2017*; *Miller, 1980*; *Schinzel and Hahn, 2008*; *Stratton et al., 2009*), but their identification amid large and diverse genetic alterations is challenging. Our results suggest the key positioning of early metazoan regulator genes at the interface of network regions from unicellular and multicellular ancestors makes cells susceptible to widespread dysregulation of transcriptional networks upon their disruption, as their alteration would uncouple the regulatory controls required for multicellularity (*Greaves, 2015*). This implicates these genes in key roles in the onset and progression of cancer and highlights them as potential gene drivers and drug targets. Our analysis of

the effect on cell line dependency after knockout of these regulators revealed that their alteration is capable of modulating cell proliferation across 364 cell lines.

Many of these interface regulators have not been significantly studied in the context of cancer, but drugs targeting these genes are currently in clinical trial for other diseases, opening the possibility for drug repurposing. LRRK2 encodes the dardarin protein, considered to be central to the aetiology of Parkinson disease (*Zimprich et al., 2004*). Two inhibitors of dardarin, DNL201 and DNL151, are currently undergoing testing in clinical trials as a means to slow down or regress neurodegenerative diseases (*Atashrazm and Dzamko, 2016*). We also identified that cell lines dependent on the UC/EM-i regulator PPRC1, peroxisome proliferator-activated receptor gamma, co-activator-related 1, were particularly susceptible to mTOR inhibitors, YK-4–279 and docetaxel. PPRC1 is an activator of mitochondrial biogenesis, a process regulated by mTOR (*Morita et al., 2013*; *Ramanathan and Schreiber, 2009*; *Schieke et al., 2006*), highlighting the use of mTOR inhibitors in cancers with aberrant mitochondrial activity. Furthermore, it suggests that YK-4–279, a binding inhibitor of oncogenic fusion proteins in Ewing's sarcoma and with encouraging pre-clinical efficiency in this cancer (*Lamha-medi-Cherradi et al., 2015*), could also be broadly effective in general for tumors with aberrant mitochondrial activity.

This study provides comprehensive evidence that both the frequency and types of mutations in genes in cancer are strongly influenced by a given gene's evolutionary age and its regulatory functions. This furthers our understanding of how a limited number of genetic alterations could promote rapid tumor development through loss of multicellular features and provides an explanation as to how the widespread convergence to common hallmark phenotypes in solid cancer may occur. As we show, this approach can be used to prioritize genes as drivers and identify possible targeted therapies, creating a novel analytical framework that will become increasingly informative as the volume and resolution of cancer genomics data continue to increase.

## Materials and methods

### Gene ages

The evolutionary ages of genes were obtained from previously published work (*Trigos et al., 2017*). Phylostratigraphy (*Domazet-Loso et al., 2007*) was used to map human genes onto a phylogenetic tree of 16 phylostrata, spanning genes found across all organisms (Phylostratum 1) to those specific to humans (Phylostratum 16). Human genes with orthologs in primitive unicellular species were assigned to older phylostrata (phylostrata 1–3) and referred to as unicellular (UC) genes, those with orthologs in early metazoan species (phylostrata 4–9) are referred to as early metazoan (EM) genes, and those assigned to phylostrata 10–16 are considered to be mammal-specific (MM) genes.

### Tumor types included in analysis

We included in our analysis all 30 solid cancer types available from TCGA GDAC for which there was CNA and point mutation data available. These were: Adrenocortical carcinoma (ACC), Bladder urothelial carcinoma (BLCA), Breast invasive carcinoma (BRCA), Cervical and endocervical cancers (CESC), Cholangiocarcinoma (CHOL), Colon adenocarcinoma (COAD), Esophageal carcinoma (ESCA), Glioblastoma multiforme (GBM), Head and Neck squamous cell carcinoma (HNSC), Kidney Chromophobe (KICH), Kidney renal clear cell carcinoma (KIRC), Kidney renal papillary cell carcinoma (KIRP), Brain Lower Grade Glioma (LGG), Liver hepatocellular carcinoma (LIHC), Lung adenocarcinoma (LUAD), Lung squamous cell carcinoma (LUSC), Ovarian serous cystadenocarcinoma (OV), Pancreatic adenocarcinoma (PAAD), Pheochromocytoma and Paraganglioma (PCPG), Prostate adenocarcinoma (PRAD), Rectum adenocarcinoma (READ), Sarcoma (SARC), Skin Cutaneous Melanoma (SKCM), Stomach adenocarcinoma (STAD), Testicular Germ Cell Tumors (TGCT), Thyroid carcinoma (THCA), Thymoma (THYM), Uterine Corpus Endometrial Carcinoma (UCEC), Uterine Carcinosarcoma (UCS), Uveal Melanoma (UVM) (*The Cancer Genome Atlas Network, 2017b*).

### Point mutation data

We obtained somatic point mutation data from the Genomic Data Commons Data Portal (https://portal.gdc.cancer.gov/) for all tumor types available. We selected the intersect of variants called by MuSE (*Fan et al., 2016*), MuTect2 (*Cibulskis et al., 2013*), VarScan2 (*Koboldt et al., 2012*) and

SomaticSniper (*Larson et al., 2012*) by The Cancer Genome Atlas. We excluded variants found in ExAC, 1000 genomes, and only included variants from canonical transcripts not found in the last exon. We also excluded genes with large number of known false positives based on previous studies (*Lawrence et al., 2013b*), namely titin, mucins, ryanodine receptors, dyneins, PCLO, cub- and sushi-domain proteins, neurexins, contactins, PARK2 and olfactory receptors.

Point mutations were classified as a synonymous mutation, missense mutation (missense variant, inframe deletion or inframe insertion), or loss-of-function (LoF) mutation (frameshift, stop-gain, splice acceptor variant and splice donor variant). Only missense mutations identified as being deleterious by SIFT and probably damaging by PolyPhen were included in our analyses.

We also obtained a list of recurrently point-mutated genes derived using MutSig2CV (*Lawrence et al., 2013b*) from Broad TCGA GDAC Firehose, using the firehose_get functionality (version 0.4.13). Genes with a q value less than 0.05 were considered significant.

## Copy-number aberration data

We obtained copy-number aberration (CNA) data from the Genomic Data Commons Data Portal (https://portal.gdc.cancer.gov/) obtained with SNP arrays (*The Cancer Genome Atlas Network, 2017a*). Only chromosome regions with at least 10 probes were considered. GenomicRanges (*Lawrence et al., 2013a*) was used to assign chromosome regions to genes. Those with positive segment means were considered to be amplifications, and the rest deletions. A gene was considered amplified or deleted only in cases where genes were covered entirely by a region with a CNA. Amplifications were assigned the maximum segment mean and deletions the minimum, and only CNAs with a segment mean in the top 0.90 quantile were included in the analysis.

We also obtained a list significantly CNAs genes derived using Gistic2 (*Mermel et al., 2011*) available from Broad TCGA GDAC Firehose, using the firehose_get functionality (version 0.4.13). Only genes with a confidence of 0.99 and that were in the CNA region were considered.

## Enrichment of recurrent point mutations and CNAs in phylostrata

We defined a gene as having recurrent point mutations if there were at least three patients across a particular tumor cohort with missense or LoF mutations in this gene. To account for the background mutation rate of the gene, only genes with a larger number of patients with missense or LoF mutations than with synonymous mutations were considered. We selected recurrent CNAs as those that were amplified or deleted genes in at least 10% of the patients. Note that these procedures were followed for each tumor type independently.

We calculated the fraction of CNA or point mutated genes in each phylostratum for each tumor type as follows:

$$Fraction\ of\ genes\ altered_{Phy_i} = \frac{N_{Recurrent_i}}{N_i}$$

where $N_{Recurrent_i}$ corresponds to the number of genes with a recurrent genetic alteration in phylostratum $i$, and $N_i$ the total number of genes in phylostratum $i$.

To account for differences in ranges between tumor types, we ranked the phylostrata by the fraction of genes altered, ranging from 1 (most altered) to 16 (least altered).

To compare the trends of recurrent with non-recurrent alterations, we calculated the fraction of altered genes in each phylostratum that were non-recurrent.

$$Fraction\ of\ non-recurrent\ genes\ altered_{Phy_i} = \frac{N_{Non-recurrent_i}}{N_{Non-recurrent_i} + N_{Recurrent_i}}$$

where $N_{Non-recurrent_i}$ corresponds to the number of genes with a non-recurrent genetic alteration in phylostratum $i$, and $N_{Recurrent_i}$ the number of genes with recurrent alterations in phylostratum $i$.

## Gene expression analysis of mutated genes

RNAseq gene expression data from the tumor samples and the corresponding normal samples were obtained from The Cancer Genome Atlas (*The Cancer Genome Atlas Network, 2015*). To determine how the mutational status of a gene affected its expression, we compared the expression of each gene with missense mutations in at least three patients or LoF mutations in at least three

patients in the tumor type cohort against their levels in patients where the gene was unmodified. One-sided Wilcoxon tests were used to determine whether a gene was significantly over or underexpressed in patients where the gene was mutated, using a p-value cut-off of 0.05. We subsequently calculated the ratio of the number of up- and downregulated genes in each phylostratum.

We performed a similar procedure to calculate the effect of point mutations in regulators over the expression of their downstream targets, comparing the level of expression of each downstream target in tumor samples with and without the regulator being mutated using two-sided Wilcoxon tests (p<0.05). In this case, however, we pooled samples across tumor types to increase power due to the small number of point mutations that occur in regulators. Specifically, we only considered regulators that were point mutated in at least three samples across tumor types, and compared the expression of their downstream targets against those of samples of the tumor types where the regulator was not mutated.

The magnitude of the downstream effect of mutating regulators was classified as low (<5% down downstream targets affected), moderate (5–20%) or high (>20%) impact. We defined the prevalence of EM genes as regulators as the ratio of the number of EM regulators over the number of UC regulators. The log10 values were calculated to normalize to zero.

## Signatures of selection in chromosomes

We defined the fraction of copy-number aberrant chromosome for each patient and chromosome as the ratio of the number of genes affected by amplifications or deletions and the total number of genes in the chromosome. To associate this chromosomal context with evolutionary ages of genes, we determined the presence or absence of genes of a specific phylostratum in the copy-number aberrant chromosome regions of each patient. The information was aggregated across patients by averaging the fraction of chromosome altered for each chromosome and phylostratum. Genes in shorter CNA regions (smaller fraction) were considered to be under stronger, focal selection, whereas CNAs in genes found in longer CNA regions (larger fraction) were considered to result from broad amplifications or deletions.

We defined focally amplified or deleted genes as those in which their chromosomal context was in the upper quantile (0.25) of the distribution of the mean fraction of copy-number aberrant chromosome across patients for each chromosome.

## Gene regulatory network analyses

We obtained a human gene regulatory network (GRN) from PathwayCommons (version 9) (*Cerami et al., 2011*; *Pathway Commons, 2017*) by selecting pairs of genes connected by an edge of the type 'control-expression-of', resulting in a directed network with 95,651 edges. We defined regulator genes as those with at least one downstream target.

We defined 'out-degree' as the number of outgoing edges of a regulator, representing the extent of its downstream regulatory network. In contrast, the 'in-degree' of a gene was defined as the number of incoming edges and it is proportional to how highly regulated the gene is. Greater out-degree/in-degree ratios indicate bias towards a higher number of outgoing edges (i.e. regulatory role), whereas smaller ratios indicate bias towards a higher number of incoming edges (i.e. target role).

We also obtained protein-protein (PPI) networks of humans from PathwayCommons (*Cerami et al., 2011*) version 9, BioGRID (*Chatr-Aryamontri et al., 2017*) version 3.4.152 and the InWeb_IM network (*Li et al., 2017*). Only nodes and edges corresponding to genes and links between two genes were considered. Since these networks are undirected, we only calculated the degree of a gene as the total number of edges associated with the gene.

## Classification of regulators by the evolutionary ages of their downstream targets

Regulators were classified as UC-t, EM-t, MM-t or UC/EM-i regulators based on the ages of their target genes, independently of the evolutionary age of the regulator. First, we calculated the percentage of downstream UC, EM and MM target genes for each regulator. A regulator was classified as UC-t if more than 2/3 of its targets were UC genes, as EM-t if more than 2/3 of its targets were EM genes, or MM-t if more than 2/3 if its targets were MM. Regulators that did not meet the above

criteria, but at least 1/10 of their target genes were UC genes or EM genes, and less than 1/10 were MM genes, were classified as UC/EM-i regulators.

Functional enrichment of UC/EM-i regulators was performed using the R package of gProfiler (*Reimand et al., 2007*) (version 0.6.4). Electronic annotations and gene sets with more than 1000 genes were excluded from the enrichment analysis.

We compared the effect of point mutations on the level of the expression of pathways they are involved in by calculating the ssGSEA scores (*Barbie et al., 2009*) using GSVA (version 1.20.0) (*Hänzelmann et al., 2013*) of KEGG pathways available in the Molecular Signatures Database from the Broad Institute (v6.2) (*Liberzon et al., 2011*; *Subramanian et al., 2005*) and performing Wilcoxon test with Benjamini-Hochberg correction. Pathways that did not include an UC/EM-i regulator or referred to generic disease pathways were excluded from the analysis. Only pathways with an adjusted p-value less than 0.05 were considered significant.

## CNA in regulators and targets

To determine whether the copy-number status of a regulator was associated with the fraction of downstream CNA targets, we calculated the fraction of downstream CNA targets for each regulator in each patient. We compared the fraction of CNA downstream target genes when regulators were CNA and CNN for each patient, and used Wilcoxon tests to determine if there were significant differences in these fractions between patients where the regulator was CNA or CNN. We only considered regulators that were CNA and CNN in at least three patients across all tumor cohorts, and had at least two targets. p-Values were corrected for multiple testing using Benjamini-Hochberg correction.

A summary fraction of CNA targets was obtained for each regulator by calculating the median across patients, and we calculated the difference in percentage of targets with CNAs by subtracting the fraction of targets when the regulator was CNA minus the fraction when the regulator was CNN.

## Gene expression analyses of CNA targets and regulators

RNAseq gene expression data from the tumor samples and the corresponding normal samples were obtained from The Cancer Genome Atlas. We evaluated the effect of CNAs on gene expression by calculating the expression fold-change between matched tumor and normal samples for each gene, and comparing the fold-changes of samples where the gene was CNA and where it was CNN using one-sided Wilcoxon tests. Only genes that were amplified or deleted in at least three samples were considered. Benjamini-Hochberg correction was used for correction for multiple testing. Note that due to the poor overlap between patient gene expression data for tumor and matched normal samples and CNA values, data presented is only of a subset of the 30 tumor types. We next calculated the percentage of significantly over or underexpressed amplified or deleted UC, EM and MM target genes, respectively. We calculated the percentage of differentially expressed CNA targets as the ratio between the number of differentially expressed CNA targets of each regulator and the total number of CNA target genes multiplied by 100. We subsequently calculated the median percentage of differentially expressed CNA targets for each regulator class in each tumor type, and Wilcoxon tests were calculated to determine significant differences.

## Cancer cell line gene knockout, mutation and IC50 values

Scores of the probability of dependency to genes across 364 solid-tumor tissue cell lines were obtained from the Avana CRISPR-Cas9 genome-scale knockout dataset generated by Project Achilles and the Cancer Dependency Map project (18Q1 version) (https://portals.broadinstitute.org/achilles; *Trigos et al., 2018*). We excluded all cell lines from haematopoietic and lymphoid tissues. A cell line was considered to be dependent on a regulator if its probability of dependency was greater than 0.95. The enrichment of a regulator class among the regulators to which cancer cell lines were dependent was determined using the odds ratio.

Mutation and CNA information of cell lines was obtained from the Cancer Cell Line Encyclopaedia (CCLE) (https://portals.broadinstitute.org/ccle; *Broad Institute, 2013*; *Broad Institute, 2018a*). Significant differences in the dependency scores of cell lines with mutated and non-mutated regulators, or amplified or copy-number normal regulators were obtained using Wilcoxon tests (p<0.05). We only considered regulators that are either amplified or point mutated in at least three cell lines.

IC50 values of cancer cell lines after treatment with 250 cancer drugs were obtained from the Genomics of Drug Sensitivity in Cancer database (version 17, release 6) (*Yang et al., 2013*) (https://www.cancerrxgene.org/; *Genomics of Drug Sensitivity in Cancer Consortium, 2016*). We calculated the Spearman correlation of IC50 values with dependency scores from the Avana CRISPR-Cas9 databases. We only considered negative correlations $< -0.25$ and with a adjusted p-values after Benjamini and Hochberg correction $<0.05$, since we were interested in cell lines with high dependency to a regulator and that showed greater drug sensitivity at lower concentrations.

## Code availability

R code to perform all analyses and generate all figures is available at https://github.com/cancer-evolution/Evolutionary-analysis-of-somatic-mutations-in-cancer (*Trigo, 2019*; copy archived at https://github.com/elifesciences-publications/Evolutionary-analysis-of-somatic-mutations-in-cancer).

## Acknowledgements

This work was supported by a Melbourne International Engagement Award (MIEA) and a Melbourne International Fee Remission Scholarship (MIFRS) from the University of Melbourne to AST and funding from the National Health and Medical Research Council of Australia (NHMRC) (APP1052904) and the Peter MacCallum Cancer Foundation to DLG, NHMRC Senior Research Fellowships to ATP, and a National Health and Medical Research Council (NHMRC) of Australia Program Grant (#1053792) and Fellowship to RBP.

## Additional information

### Funding

| Funder | Author |
|---|---|
| University of Melbourne | Anna S Trigos |
| National Health and Medical Research Council | Richard B Pearson<br>Anthony T Papenfuss<br>David L Goode |
| The Peter MacCallum Cancer Centre Foundation | David L Goode |

The funders had no role in study design, data collection and interpretation, or the decision to submit the work for publication.

### Author contributions

Anna S Trigos, Conceptualization, Formal analysis, Investigation, Visualization, Methodology, Writing—original draft, Writing—review and editing; Richard B Pearson, Supervision, Writing—review and editing; Anthony T Papenfuss, Supervision, Provided advise on methodologies and project design; David L Goode, Conceptualization, Supervision, Methodology, Writing—review and editing

### Author ORCIDs

Anna S Trigos (iD) http://orcid.org/0000-0002-5915-2952
David L Goode (iD) http://orcid.org/0000-0002-3277-6562

### Decision letter and Author response

Decision letter https://doi.org/10.7554/eLife.40947.052
Author response https://doi.org/10.7554/eLife.40947.053

## Additional files

### Supplementary files

• Supplementary file 1. Point mutated genes. Genes marked as 'Frequent' were included in the analysis.
DOI: https://doi.org/10.7554/eLife.40947.037

• Supplementary file 2. Ratio of out-degree and in-degree per age and mutation type.
DOI: https://doi.org/10.7554/eLife.40947.038

• Supplementary file 3. Regulator classification.
DOI: https://doi.org/10.7554/eLife.40947.039

• Supplementary file 4. Functional enrichment analysis of UC/EM-i regulators using gprofileR.
DOI: https://doi.org/10.7554/eLife.40947.040

• Supplementary file 5. Significance of change of pathway activity levels after point mutation of UC/EM-i regulators.
DOI: https://doi.org/10.7554/eLife.40947.041

• Supplementary file 6. Difference in median dependency between cell lines with regulator mutated and non-mutated.
DOI: https://doi.org/10.7554/eLife.40947.042

• Transparent reporting form
DOI: https://doi.org/10.7554/eLife.40947.043

### Data availability

All data used during this study was obtained from the public databases indicated in the manuscript. Results generated during this study are included as supporting files.

The following previously published datasets were used:

| Author(s) | Year | Dataset title | Dataset URL | Database and Identifier |
|---|---|---|---|---|
| Broad Institute | 2018 | MSigDB | http://software.broadinstitute.org/gsea/msigdb/genesets.jsp?collection=CP:KEGG | Molecular Signatures Database, CP:KEGG |

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

# Appendix 1

DOI: https://doi.org/10.7554/eLife.40947.044

We repeated our analysis using two additional databases, TRRUST, a transcription factor-target interaction database obtained by text-mining (version 2, http://www.grnpedia.org/trrust/) (*Han et al., 2018*) and RegNetwork, a compendium of multiple regulatory network databases (http://www.regnetworkweb.org/home.jsp) (*Liu et al., 2015*).

As with the GRN from PathwayCommons, we only included in our analysis edges for which gene ages were available for both the regulator and target, resulting in the TRRUST network having 7222 edges, with 719 regulators and 2187 targets, and the RegNetwork having 104,626 edges, with 1254 regulators and 14,677 targets.

Similar to the GRN from PathwayCommons, the GRN from both TRRUST and RegNetwork were dominated by regulators of EM origin (66.48% and 58.69, respectively, enriched $p=8.25\times10^{-10}$ and $p=2.38\times10^{-7}$) (*Appendix 1—figure 1A, D*). EM genes also had a higher out-degree (11.19 compared to 8.38 for UC genes and 4.41 for MM genes in TRRUST, and 100.21 compared to 65.27 for UC genes and 28.99 for MM genes in RegNetwork) and in-degree (3.68 compared to 2.72 for UC genes and 3.20 for MM genes in TRRUST, and 7.58 compared to 7.29 for UC genes and 4.46 for MM genes) (*Appendix 1—figure 1B,E*). Finally, 72.10% and 63.95% of the most connected regulators (upperquantile of the distribution of out-degree) were EM genes in TRRUST and RegNetwork, respectively. With respect to mutations in regulators and targets, we also found that regulators were preferentially affected by point mutations (Wilcoxon test $p=1.44\times10^{-5}$ in TRRUST and $9.70\times10^{-6}$ in RegNetwork), and that targets showed a preference for CNAs (Wilcoxon test $p=9.12\times10^{-7}$ in TRRUST and $9.12\times10^{-7}$ in RegNetwork) (*Appendix 1—figure 1C, F*). Additionally, for genes with both a regulatory and target role, we found that EM genes with recurrent point mutations had a stronger regulatory role than target role in both databases (*Appendix 1—figure 2A, B*).

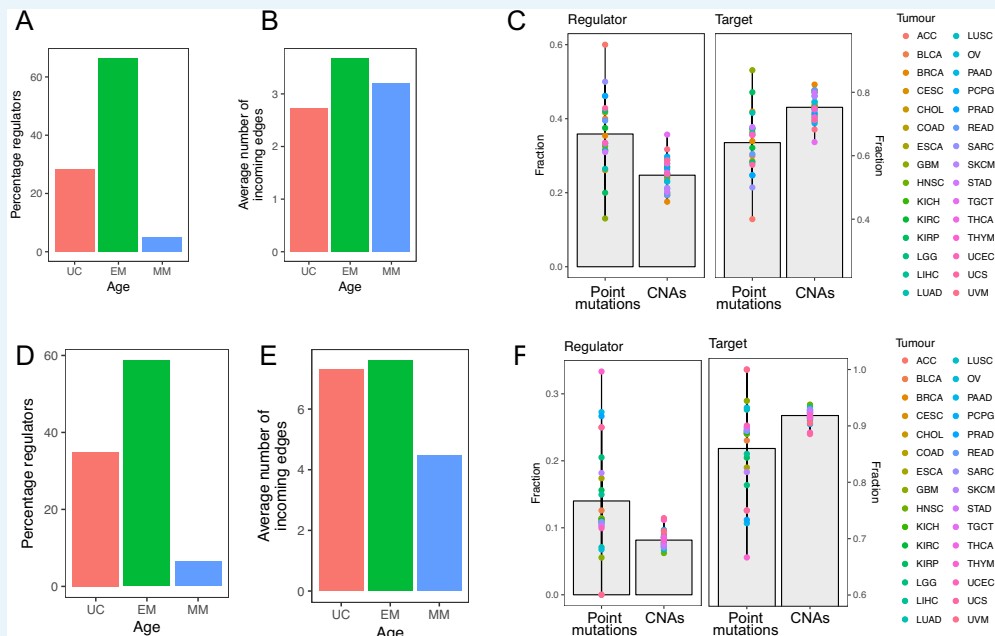

**Appendix 1—figure 1.** Results obtained using the TRRUST and RegNetwork databases pertaining to network composition and distribution of mutations in regulators and targets. The top

row corresponds to results obtained with TRRUST, and the bottom row those obtained with RegNetwork. The results are largely consistent with those obtained with the GRN from PathwayCommons.

DOI: https://doi.org/10.7554/eLife.40947.045

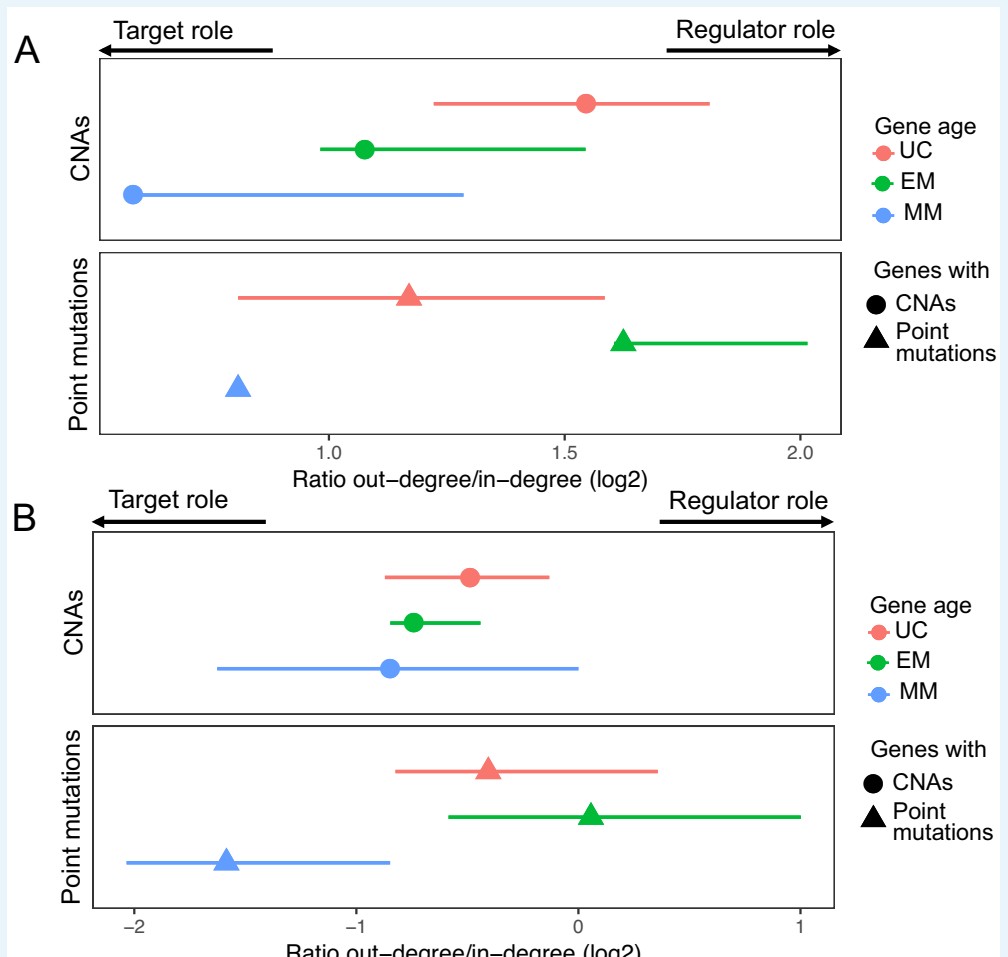

**Appendix 1—figure 2.** Results obtained using the TRRUST and RegNetwork databases related to the regulatory and target roles of genes with point mutations and CNAs. In both the TRRUST (A) and RegNetwork (B), EM genes with point mutations have a stronger regulatory role (high out-degree/in-degree ratio) than UC and MM genes with point mutations. In contrast, genes affected by CNAs have a mostly target role.

DOI: https://doi.org/10.7554/eLife.40947.046

Reclassification of regulators into UC-t, UC/EM-i and EM-t based on the same criteria using each database individually resulted in similar classifications of regulators. Of note, many regulators were only available in the PathwayCommons database, so we could not validate in the other two databases. Of the ones available in the three databases, 83.80% of UC/EM-i regulators were also classified as such in at least one other database (**Appendix 1—figure 3**). UC/EM-i regulators were also enriched in EM genes in both databases (p=$9.64\times10^{-3}$ in TRRUST and $3.38 \times 10^{-7}$ in RegNetwork). Commonly recurrent point mutations across tumors were preferentially found in UC/EM-i regulators of TRRUST and RegNetwork (66.67% in both) (**Appendix 1—figure 4A,C**). The preferential CNAs of downstream targets of UC-t and EM-t regulators, but not of UC/EM-i regulators, was also observed when using both databases

(*Appendix 1—figure 4B,D*) (p=0.0055 comparing UC-t and UC/EM-i, and 0.19 comparing EM-t and UC/EM-i using TRRUST, and p=2.21×10$^{-11}$ comparing UC-t and UC/EM-i, and 5.66 × 10$^{-21}$ comparing EM-t and UC/EM-i using RegNetwork).

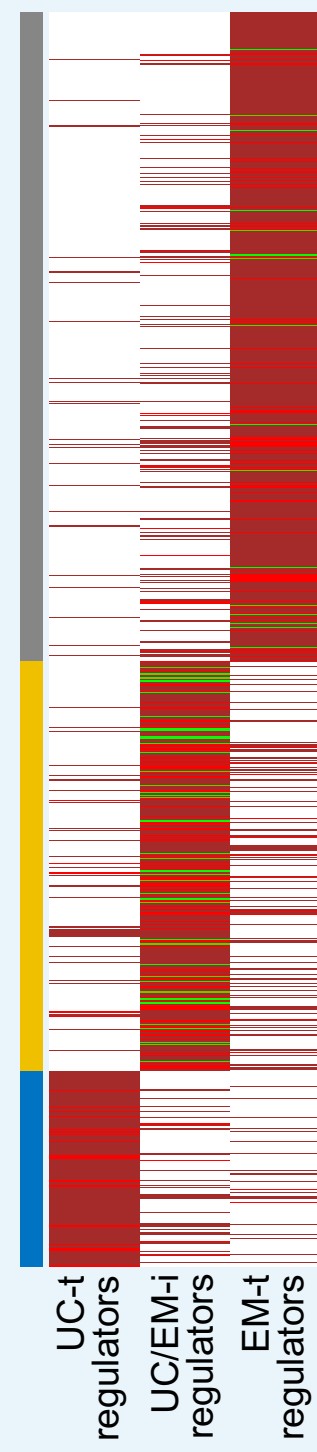

**Appendix 1—figure 3.** Concordance of regulator classification across databases. The classification shown on the left corresponds to that obtained with GRN from PathwayCommons. Many regulators are only found in this databases (large sections of dark red). Although there is some variability in the classification of regulators across databases, UC/EM-i regulators are more likely to be classified as such by the three databases (green lines).

DOI: https://doi.org/10.7554/eLife.40947.047

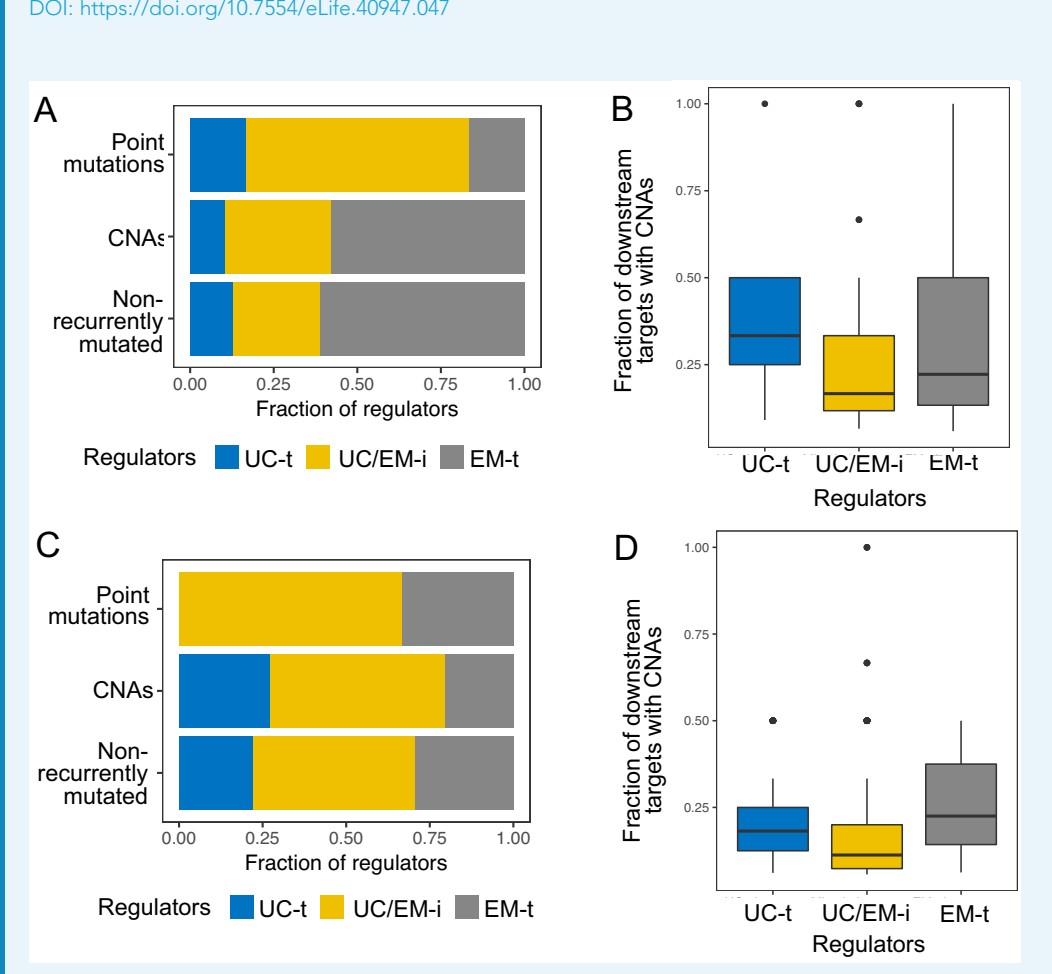

**Appendix 1—figure 4.** Results obtained using the TRRUST and RegNetwork databases pertaining to point mutations and CNAs in different classes of regulators. The top row corresponds to results obtained with TRRUST, and the bottom row those obtained with RegNetwork. The results are largely consistent with those obtained with the GRN from PathwayCommons.

DOI: https://doi.org/10.7554/eLife.40947.048

