## [Decision Letter]

Thank you for submitting your article "Somatic mutations in early metazoan genes disrupt regulatory links between unicellular and multicellular genes in cancer" for consideration by *eLife*. Your article has been reviewed by three peer reviewers, and the evaluation has been overseen by a Reviewing Editor and Aviv Regev as the Senior Editor. The following individual involved in review of your submission has agreed to reveal his identity: Kevin Litchfield (Reviewer #1).

The reviewers have discussed the reviews with one another and the Reviewing Editor has drafted this decision to help you prepare a revised submission.

Summary:

The paper adds to the analysis of the relationship between evolutionary age of genes and their possible role in cancer. A representative set of genes are analysed in terms of accumulated point mutations and copy number changes in seven solid tumours from TCGA. The results are then discussed in terms of the relation between gene age, type of mutations and functional role of the genes as master in gene regulation at different levels. The difference between mutations and CNVs is also interpreted in terms of a potential differential role in gene regulation.

Essential revisions:

There are questions about the use of a more complete TCGA dataset and selection of the initial gene set that affects the scope of the study and results. They have to be properly addressed.

There is also an important question about the relation between the possible positive selection on point mutations in early EM regulators that requires specific considerations, as well as a number of other comment and criticisms regarding clarity and interpretation of the results.

In more detail:

Data completeness:

1) TCGA currently comprises of data from a large number of tumour types. The authors have analysed data from only seven tumour types? Why have the authors conducted the analysis in only a subset of the available solid tumour types in TCGA? Several solid tumour types have been missed (e.g. renal, melanoma, testicular, pancreatic, etc) with no apparent explanation. Please could the authors either extend the analysis to the full TCGA solid tumour dataset, or clarify why only a subset of available data has been utilised.

2) The gene filtering strategy (subsection “Enrichment of recurrent point mutations and CNAs in phylostrata”) raises some concern – only taking genes with more missense+LoF than synonymous mutations may miss some important driver events (e.g. by chance a long gene may carry a number of synonymous mutations, and then missense mutations only in hotspot domains (KRAS G12D, V600E, etc)). By chance drivers may be excluded. Could the authors confirm whether this approach was done on a per tumour type basis, or across all histologies combined? The latter would be particularly concerning. As a minimum the authors should provide a supplementary table showing this data and the genes excluded, so readers can confirm what is being excluded. A better approach instead could be to take only genes significant in MutSigCV analysis for each cancer type (which are already available for all tumor types in the Broad Firehose repository). The mutsigCV algorithm implements a similar method but in a much comprehensive way.

Presentation:

1) Some sections of the text (particularly Results section “Point mutations disrupt the regulation between UC and EM Genes”) should be cut down and streamlined, so the salient (and significant) results are better highlighted, and other non-significant descriptive text removed. For example endless "x out of y" results are quoted, e.g. (4/6 and 5/6 regulators). These proportions are stated with no associated p-values, so it is somewhat unclear to the reader if these significant results or not?

2) The authors have tried to link transcriptional states with mutation states of genes such as gene expression down-regulation has been linked to missense or LoF mutations. This claim needs to be supported with some evidence. It will be particularly helpful for the reader if the authors are able to give more biological context in their analysis. I think this study will definitely benefit from more examples of specific UC/EM-i genes and the pathways they regulate in tumorigenesis.

Positive selection of EM regulators:

The authors propose a model of point mutations in early EM regulators being key drivers under positive selection in cancer (paragraph five “Point mutations disrupt the regulation between UC and EM genes”). This point needs to be further substantiated, by showing these mutated genes are indeed enriched as clonal driver events. Can the authors show the variant allele frequencies for these EM regulator mutations are higher than average, or higher than other groups? This would support their role as drivers and presence in a high proportion of cancer cells.

---

## [Author Response]

Essential revisions:There are questions about the use of a more complete TCGA dataset and selection of the initial gene set that affects the scope of the study and results. They have to be properly addressed.There is also an important question about the relation between the possible positive selection on point mutations in early EM regulators that requires specific considerations, as well as a number of other comment and criticisms regarding clarity and interpretation of the results.In more detail:Data completeness:1) TCGA currently comprises of data from a large number of tumour types. The authors have analysed data from only seven tumour types? Why have the authors conducted the analysis in only a subset of the available solid tumour types in TCGA? Several solid tumour types have been missed (e.g. renal, melanoma, testicular, pancreatic, etc) with no apparent explanation. Please could the authors either extend the analysis to the full TCGA solid tumour dataset, or clarify why only a subset of available data has been utilised.

The analysis was carried out in the 7 tumour types that we were used in a previous study (Trigos et al., 2017), since the current manuscript represents a continuation of that work. However, we have now updated all of our analysis with data of the 30 tumour types available in TCGA for which there is both mutation and CNA data (no CNA data is available for mesothelioma, and therefore this cancer type was not included in the analysis). Our results with the new dataset are consistent with the previous results, which strengthen our claims.

2) The gene filtering strategy (subsection “Enrichment of recurrent point mutations and CNAs in phylostrata”) raises some concern – only taking genes with more missense+LoF than synonymous mutations may miss some important driver events (e.g. by chance a long gene may carry a number of synonymous mutations, and then missense mutations only in hotspot domains (KRAS G12D, V600E, etc)). By chance drivers may be excluded. Could the authors confirm whether this approach was done on a per tumour type basis, or across all histologies combined? The latter would be particularly concerning. As a minimum the authors should provide a supplementary table showing this data and the genes excluded, so readers can confirm what is being excluded. A better approach instead could be to take only genes significant in MutSigCV analysis for each cancer type (which are already available for all tumor types in the Broad Firehose repository). The mutsigCV algorithm implements a similar method but in a much comprehensive way.

The filtering based on the ratio of missense or LoF mutation with synonymous mutations was done on a per tumour type basis. We have included a supplementary table (Supplementary File 1) with the genes that survived this filtering in each tumour type and were used in the analyses presented in the paper.

We also included all relevant analyses using genes identified as significantly mutated by MutSig2CV using the files available in the Firehose repository. The results are consistent with the ones obtained filtering solely by the ratio with the number of synonymous mutations (compare Figure 1 panels A and E with Figure 1—figure supplement 4, Figure 2 panels E and F with Figure 2—figure supplement 5 and 6, and Figure 3 panels B and C with Figure 3—figure supplement 1).

Presentation:1) Some sections of the text (particularly Results section Point mutations disrupt the regulation between UC and EM Genes) should be cut down and streamlined, so the salient (and significant) results are better highlighted, and other non-significant descriptive text removed. For example endless "x out of y" results are quoted, e.g. (4/6 and 5/6 regulators). These proportions are stated with no associated p-values, so it is somewhat unclear to the reader if these significant results or not?

This Results section has been streamlined to improve clarity and better highlight significant results. We have reduced the number of categories to define the level of change of expression of target genes from 3 (high, moderate, low impact) to 2 (high or low impact). The inclusion of 23 additional tumour types as well as the reduction in the number of categories has also increased the number of regulators and point mutated genes in the set. We found that 12/16 regulators affected by high-impact point mutations are UC/EM-i genes (75%), whereas this proportion is considerably smaller for low-impact mutations (22/51, 43.14%). This was significant by a two-proportions Z-test (p = 0.026), indicating that high impact mutations affect a higher proportion of UC/EM-i regulators than low impact mutations.

2) The authors have tried to link transcriptional states with mutation states of genes such as gene expression down-regulation has been linked to missense or LoF mutations. This claim needs to be supported with some evidence. It will be particularly helpful for the reader if the authors are able to give more biological context in their analysis. I think this study will definitely benefit from more examples of specific UC/EM-i genes and the pathways they regulate in tumorigenesis.

The evidence for the changes in gene expression associated to mutations is presented in Figure 1C, where a higher percentage of EM genes with point mutations are differentially expressed than UC or EM genes. This analysis includes data from 30 tumour types and 9675 missense mutations and 2367 LoF mutations. We show EM genes are more likely to be differentially expressed when point mutated than are UC and MM genes across most of the cancers we assessed.

To provide more biological context to the reader, we have included a functional enrichment analysis of the UC/EM-i regulators and also evaluated the association between the expression level of multiple pathways relevant to cancer and point mutations in UC/EM-i regulators in subsection “Point mutations disrupt the regulation between UC and EM genes”. This uncovered many differences in key pathways are linked to mutations in certain regulators, the implications of which are discussed in the third paragraph of the Discussion section.

We interpret these findings as support for the direct disruption of activity of key EM genes across cancers by point mutations.

Positive selection of EM regulators:The authors propose a model of point mutations in early EM regulators being key drivers under positive selection in cancer (paragraph five “Point mutations disrupt the regulation between UC and EM genes”). This point needs to be further substantiated, by showing these mutated genes are indeed enriched as clonal driver events. Can the authors show the variant allele frequencies for these EM regulator mutations are higher than average, or higher than other groups? This would support their role as drivers and presence in a high proportion of cancer cells.

We present multiple sources of evidence that point towards mutations in EM regulators as key drivers. We show that recurrent mutations in EM genes are the most frequent (Figure 1 – panel B), that genes with point mutations affect genes that have a strong regulatory role (Figure 2 – panel F), and we further show in Figure 3 that this effect is strongest for UC/EM-i regulators of EM origin.

Our model in particular emphasizes the importance of UC/EM-i regulators that sit at the interface between UC and EM regions of the regulatory network. This set we would expect plays a very important role in tumourigenesis when mutated and thus comprises a significant proportion of early driver mutations in many tumours.

To assess this, we turned to a study that derived clonality and driver status for individual patients using machine learning applied to multi-region sequencing data from cohorts outside TCGA (Caravagna et al., 2018). The overwhelming majority (91%) of non-small-cell lung cancers from the TRACERx study (Jamal-Hanjani et al., 2017) and the Yates cohort of breast cancers (71%) (Yates et al., 2015) have at least one UC/EM-i regulators labelled as a clonal driver event. Thus, the number of driver and clonal point mutated regulators in individual patients shows a high prevalence of UC/EM-i regulators as clonal drivers across lung cancer and breast cancer patients, supporting our claims of UC/EM-i regulators playing a key role in cancer development. These results have been added to the manuscript (second paragraph of subsection “UC/EM-i regulators are important drivers of tumourigenesis and influence drug sensitivity” of the Results) and are displayed in the new Figure 5—figure supplement 1.

Similar data on sub-clonal populations have been produced for selected TCGA tumour types, however these results are not available with the degree of detailed required. Applying clonality prediction algorithms ourselves to all 30 TCGA types used here would be highly computationally intensive and it would be very time consuming to go through the results.

Thus we turned to analysis of VAFs in our key regulator classes, as this reviewer suggests. We calculated the VAF of the variants identified by all four somatic SNV detection methods used by the TCGA consortium (Muse, Mutect, SomaticSniper and Varscan). Given that the VAF substantially differed between variant callers, we analysed them individually and excluded all genes that were identified as being CNA as copy-number changes will skew the VAF. Since differences in purity affect the range of VAFs between patients, we scaled the VAF of each patient into quantiles to avoid having to select an a priori VAF cutoff. We looked for a shift to higher quantiles in UC/EM-i regulators, under the expectation clonal driver mutations would have higher VAF than the majority of other mutations in a given tumour. To boost power only patients with at least 20 mutated genes were considered.

Regardless of variant caller, we consistently see a bimodal distribution of VAFs in UC/EM-i regulators with a noticeable peak at VAF >50% (yellow histograms in Author response image 1). This suggests a substantial proportion of UC/EM-i regulator mutations are clonal. We see many high VAFs in other regulator types as well, consistent with the notion that point mutations in regulators are generally important drivers of cancer.

Finally, comparing the percentage of significantly and non-significantly recurrently mutated regulators based on the output from MutSig2CV, we see UC/EM-i regulators comprise a higher percentage of significant mutated genes than non-significantly mutated genes in 27 out of the 30 tumour types, indicating the preference for recurrent mutations in these regulators. See Author response image 2.

**Author response image 2. respfig2:** 

Collectively, these new analyses point to coding mutations in UC/EM interface regulators as being early, selected driver events across a range of tumours.

However, as it can be difficult to unambiguously assign a selective advantage to specific somatic mutations in individual cases, we have reworded the relevant passage (now last paragraph of subsection “Point mutations disrupt the regulation between UC and EM genes” of the results in the revised manuscript) to refer to the recurrence of point mutations in these regulators, rather than selection.